



# An investigation on the origin of regional spring time ozone episodes in the Western Mediterranean and Central Europe

Pavlos Kalabokas[1,2], Jens Hjorth[2], Gilles Foret[3], Gaëlle Dufour[3], Maxim Eremenko[3], Guillaume Siour[3], Juan Cuesta[3], Matthias Beekmann[3]

[1]Academy of Athens, Research Center for Atmospheric Physics and Climatology, Athens, Greece

[2]European Commission, Joint Research Centre (JRC), Institute for Environment and Sustainability (IES), Air and Climate Unit, I - 21027 Ispra (VA), Italy

[3]Laboratoire Inter-universitaire des Systèmes Atmosphériques (LISA), Universités Paris-Est Créteil et Paris Diderot, CNRS, Créteil, France

*Correspondence to*: Pavlos Kalabokas (pkalabokas@academyofathens.gr)

**Abstract.** For the identification of regional spring time ozone episodes, rural EMEP ozone measurements from countries surrounding the Western Mediterranean (Spain, France, Switzerland, Italy, Malta) have been examined with emphasis on periods of high ozone concentrations, according to the daily variation of the afternoon (12:00 – 18:00) ozone values. For two selected high ozone episodes in April and May 2008, composite NCEP/NCAR reanalysis maps of various meteorological parameters and/or their anomalies (geopotential height, specific humidity, vertical wind velocity omega, vector wind speed and temperature) at various tropospheric pressure levels have been examined together with the corresponding satellite IASI ozone measurements (at 3 and 10 km), CHIMERE simulations, vertical ozone soundings and HYSPLIT back trajectories. The results show that high ozone values are detected in several countries simultaneously over several days. Also, the examined spring ozone episodes over the Western Mediterranean and in Central Europe are linked to synoptic meteorological conditions very similar to those recently observed in summertime ozone episodes over the Eastern Mediterranean (Kalabokas et al., ACP, 2013; Doche et al., ACP, 2014; Kalabokas et al., TellusB, 2015), where the transport of tropospheric ozone-rich air masses through atmospheric subsidence influences significantly the boundary layer and surface ozone concentrations. In particular, the geographic areas with observed tropospheric subsidence seem to be the transition regions between high pressure and low pressure systems. Over these areas, strong gradients of geopotential height and temperature are observed, together with high positive omega vertical wind velocity (downward transport) and low specific humidity (dry conditions), at all examined pressure levels below the altitude of 500 hPa pressure level. During the surface ozone episodes IASI satellite measurements show extended areas of high ozone in the lower and upper troposphere over the low pressure system areas, adjacent to the anticyclones, which influence significantly the boundary layer and surface ozone concentrations within the anticyclones by subsidence and advection in addition to the photochemically produced ozone there, resulting in exceedances of the 60 ppb standard.





## 1 Introduction

Surface ozone is a pollutant harmful to both human health and vegetation (Levy et al., 2001; Fuhrer, 2009). Further, in the upper troposphere ozone acts as a powerful greenhouse gas (IPCC, 2007). The concentrations of ozone throughout the troposphere depend on the meteorological conditions driving vertical and horizontal transport and on photochemical ozone production from its precursors, nitrogen oxides ($NO_x$) volatile organic compounds (VOCs) and carbon monoxide (Delmas et al., 2005; Seinfeld and Pandis, 2006; Monks et al., 2015).

The European network of surface ozone monitoring stations shows persistently exceedances of the European long-term target value for protection of human health, under anticyclonic synoptic meteorological conditions during the warm season in southern and central Europe (EEA, 2015). The Mediterranean area is particularly exposed to ozone pollution because of the combination of the specific meteorological conditions prevailing during spring and summer and the regional air pollutant emissions. Data collected from air pollution monitoring stations in combination with results of measurement campaigns show that ozone concentrations in the Mediterranean Basin are relatively high: Lelieveld et al. (2002) found that summer ozone concentrations over the Mediterranean are a factor of 2.5–3 higher than in the hemispheric background troposphere, in the boundary layer and up to 4 km altitude. Rural stations in continental Greece, Italy, Malta and eastern Spain report summer average ozone values of about 60–70 ppbv, significantly higher than values in Northern and Western Europe (Bonasoni et al., 2000; Kalabokas et al., 2000; Millan et al., 2000; Kourtidis et al., 2002; Kouvarakis et al., 2002; Nolle et al., 2002; Kalabokas and Repapis, 2004; Paoletti, 2006; Sánchez et al., 2008; Schürmann et al., 2009; Velchev et al., 2011, Kalabokas et al., 2008; Kleanthous et al., 2014; Cristofanelli et al., 2015). Results from 3-D chemistry transport models also suggest that ozone concentrations are higher than for the rest of Europe (e.g. Johnson et al., 2001). High ozone values in the Mediterranean are typical not only for ground level measurements, but for the entire boundary layer as well as the entire lower troposphere (Millan et al., 1997, 2000; Kalabokas et al., 2007).

The Mediterranean climate with frequent anticylonic, clear sky conditions in spring and summer favours photochemical ozone formation in the troposphere. Furthermore, Mediterranean tropospheric ozone levels are influenced by long-range transport of ozone and its precursors from Europe, Asia and even North America as well as emissions of precursors from sources around the Basin, particularly the large cities (Lelieveld et al., 2002, Gerasopoulos, 2005, Safieddine et al., 2014). Also natural VOC emissions in the area have been found to be important ozone precursors (Richards et al., 2013). Studies in the Western part of the Mediterranean highlight the role of the breeze circulation in accumulation of high ozone levels (Millan et al., 1997, 2000).

Anticyclones are generally linked to atmospheric subsidence, which seems to be particularly important over the Eastern Mediterranean as a cause of elevated ozone concentrations. During the summer period, the Mediterranean area is directly under the descending branch of the Hadley circulation, caused by deep convection in the tropics (Lelieveld, 2009). However a main reason for the strong subsidence observed in the Mediterranean Basin appears to be an impact of the Indian monsoon, inducing a Rossby wave that by the interaction with the midlatitude westerlies produces adiabatic descent in the area (Rodwell and Hoskins, 1996, 2001; Tyrlis et al., 2013). In general, in the Eastern Mediterranean strong deep subsidence in the lower troposphere influencing the boundary layer has been documented, based essentially on the analysis of MOZAIC vertical ozone profiles as





well as surface ozone and satellite measurements (Kalabokas et al., 2007; Kalabokas et al., 2008; Eremenko et
al, 2008; Foret et al., 2009; Liu et al., 2009; Kalabokas et al., 2013; Doche et al., 2014; Kalabokas et al., 2015).
Also data analysis based on large-scale atmospheric modeling studies (Li et al, 2001; Richards et al., 2013; Zanis
et al., 2014; Safieddine et al., 2014; Tyrlis et al., 2014) shows the importance of vertical transport of ozone in the
Mediterranean area, particularly in its eastern part.
Recent research based on vertical MOZAIC ozone profiles (Kalabokas et al., 2015) suggests that during days
with the highest ozone levels for both the free lower troposphere (1.5-5 km) and the boundary layer (0-1.5 km)
over the middle-eastern airports of Cairo and Tel-Aviv there are extended regions of strong subsidence in the
Eastern Mediterranean but also in Eastern and Northern Europe and over these regions the atmosphere is dryer
than average.
The influence of tropospheric transport of ozone at hemispheric scale, on surface ozone concentrations, is an
issue of potential importance to ozone abatement policies. Vertical transport of ozone is of particular relevance
to the ozone transport over long distances in the troposphere because the lifetime of ozone in the free
troposphere is longer and the transport times are typically shorter than in the boundary layer (HTAP, 2010). The
impact on surface ozone of the long range transport of ozone has been extensively investigated in the US, where
a clear difference has been observed between the eastern part, where surface ozone concentrations have
decreased very significantly, and the western part, where most sites do not show such a trend, apparently due to
the effects of transport (Cooper et al., 2012). Studies based on observations and trajectory modelling indicate
that transport above the boundary layer and entrainment of ozone from the free troposphere has an important
impact on surface ozone concentrations at several sites in the western US (Cooper et al., 2011; Langford et al.,
2015). Also a changing seasonal cycle of ozone, with a tendency towards a maximum earlier in the year, has
been tentatively explained by the influence of atmospheric transport patterns combined with a change in the
temporal and spatial emissions of ozone precursors (Parrish et al., 2013).
The importance of hemispheric transport for Mediterranean air pollution in particular has previously been
highlighted in the modelling study by Stohl et al. (2002) of intercontinental transport of air pollutants, who found
the highest concentrations of a passive tracer for North American emissions in the Mediterranean Basin. The
modelling studies of Richard et al. (2013) and Safieddine et al. (2014) both come to the conclusion that the
emission sources within the Mediterranean area have a dominating influence on surface ozone while remote
sources, e.g. in Asia and North America, are more important than local sources for ozone concentrations at
higher altitudes (above 700 hPa according to Richards et al. 2013). The radiative (climate) impact of ozone
depends mainly on the concentrations above the boundary layer. Relatively high ozone concentrations are
reported in the Central Mediterranean also during the winter-early spring period (Nolle et al., 2002) while, in an
extended observational study in the Western Mediterranean basin a converging trend between the background
rural ozone values and the urban ozone values was reported (Sicard et al., 2013).
Most studies of Mediterranean ozone have been focused on the Eastern part of the Mediterranean basin where
the influence of downwards transport appears to be more important than it is in the Central and Western part,
although also at this part of the Mediterranean very frequently anticyclonic conditions dominate in spring and
summer months. The mechanisms governing ozone levels in the Western and Central Mediterranean appear to



need further clarification. For example, in Kalabokas et al. (2008) as well as in the more recent study by Zanis et al. (2014) it was found that rural background ozone monitoring stations in the Western and Central Mediterranean show climatological spring time ozone maximum in April-May while the corresponding rural background ozone Eastern Mediterranean stations, have their climatological maxima in July-August. Also, in Eastern Greece and the Aegean Sea the summer rural background ozone levels are systematically higher than the corresponding ones observed in the Central Mediterranean (Malta) and the Eastern Mediterranean (Cyprus). The above is contrary to the predictions of a model simulation that suggested a rather uniform ozone distribution across the basin (Zanis et al., 2014). Understanding the origin of high ozone levels during spring time is particularly challenging because this is a period where photochemical formation in the troposphere is increasing due to rising sun intensity, but also stratospheric intrusions may be of relevance as these have been found to have a maximum in Southern Europe in spring/early summer (Beekmann et al., 1994; Monks, 2000).

 The focus of the present study thus is to improve the understanding of ozone behaviour over the Western Mediterranean and the surrounding area to the north part of the basin, towards central Europe in the springtime. In particular, it aims at investigating to which extend surface ozone concentrations during the high ozone episodes are influenced by entrainment of ozone rich air and how much the episodes depend on ozone formation in the boundary layer. Further, this study investigates the factors controlling ozone distribution in the free troposphere, particularly transport within the troposphere, stratosphere-troposphere exchange and photochemical formation in the free troposphere.  Two episodes in late April and early May have been selected for a detailed analysis. We analyze the two spring time episodes of high surface ozone concentrations over the Western Mediterranean and the surrounding area using a comprehensive combination of surface observations, Infrared Atmospheric Sounding Interferometer (IASI) satellite observations, meteorological maps, back trajectories and regional air quality modelling to understand the principal mechanisms contributing to these events.

## 2    Data and methodology

The following data will be used in the analysis:

1) Ozone measurements from the European Monitoring and Evaluation Programme (EMEP) and European Environment Agency (EEA) air pollution network

The surface ozone measurements during spring 2008 (March-May) of some EMEP rural ozone stations surrounding the western Mediterranean basin [Spain (ES10), France (FR10, FR14), Switzerland (CH02, CH04), Italy (IT01, IT04), Malta (MT01)]  have been analyzed with emphasis on periods of high ozone, focused mostly on the daily variation of the daily mean afternoon (12:00 – 18:00) ozone  values. In addition, during the selected spring ozone episodes the measurements of the EEA air pollution network are also taken into account for the analysis.

2) Composite NCEP/NCAR Reanalysis maps

Composite or daily NCEP/NCAR reanalysis meteorological maps covering Europe and N. Africa and corresponding to periods of high ozone have been plotted for the following meteorological parameters: Geopotential height, specific humidity anomaly, vertical wind velocity Omega (and anomaly), vector wind speed



and temperature anomaly. The examination was focused on the tropospheric pressure levels at 850, 700 and
500hPa levels (for space limitations mainly the 850hPa charts are presented). The charts are based on grids of
2.5×2.5 degrees, following the procedure of Kalnay et al. (1996).
3). HYSPLIT back trajectories
Six-day back trajectories were calculated with end points at 50, 500 and 1500 meter altitude, using the NOAA
HYSPLIT model with GDAS meteorological data (Draxler and Rolph, 2015) for the EMEP stations and the days
of the selected ozone episodes. The GDAS data have a horizontal resolution of 1 degree and 23 vertical levels
between 1000 and 20 hPa.
4). Satellite IASI ozone measurements
Satellite observations provide interesting possibilities to support the analysis of ground measurements as well as
modeling simulations. Indeed, during the last decade, satellite observations of tropospheric ozone have been
developed and have become more and more precise (e.g. Fishman et al., 2003; Liu et al., 2005; Coheur et al.,
2005; Worden et al., 2007; Eremenko et al., 2008). These observations are now able to complement in situ
observations, offering wide spatial coverage and good horizontal resolution.
The IASI instrument (Clerbaux et al., 2009), on board the MetOp-A platform since 19 October 2006, is a nadir-
viewing Fourier transform spectrometer operating in the thermal infrared between 645 and 2760 cm$^{-1}$ with an
apodized spectral resolution of 0.5 cm$^{-1}$. The IASI field of view is composed of a $2 \times 2$ matrix of pixels with a
diameter at nadir of 12 km each. IASI scans the atmosphere with a swath width of 2200 km, allowing the
monitoring of atmospheric composition twice a day at any (cloud-free) location. The spectral coverage and the
radiometric and spectral performances of IASI allow this instrument to measure the global distribution of several
important atmospheric trace gases (e.g. Boynard et al., 2009; George et al., 2009; Clarisse et al., 2011). As in
Doche et al. (2014), IASI data at 3 and 10 km height are used here for analysis. These levels are representative
for the lower and upper troposphere respectively.   However, due to the limited vertical sensitivity and resolution
of IASI, ozone concentrations retrieved at 3 km describe the ozone variability from roughly 2 to 8 km, and ozone
concentrations retrieved at 10 km describe the ozone variability from 5 to 14 km  (Dufour et al., 2010). Despite
this overlapping, recent studies show that uncorrelated information from the lower and the upper troposphere can
be derived from IASI (Dufour et al., 2010, 2012, 2015).
5) Vertical ozone soundings
Further information about the vertical ozone distribution is obtained from the ozone soundings made from the
site of Payerne, Switzerland, Uccle, Belgium and Hohenpeissenberg, Germany. These are regularly carried out
with balloon launches starting at 11 AM UTC using ECC (Electrochemical Concentration Cell) ozonesondes in
Payerne and Uccle and Brewer-Mast ozonesondes in Hohenpeissenberg. The data were downloaded from the
website of the World Ozone and Ultraviolet Data Centre (WOUDC, 2015), uncertainties on the measured ozone
concentrations for ECC ozonesondes are between 5 and 10% (Rene Stübi, personal communication).
6). Regional air quality simulations





The CHIMERE model (Menut et al, 2013) is a state-of-the-art model widely used for pollution and air quality
studies (Rouil et al., 2009; Beekmann and Vautard, 2010). For the purpose of this study, we have used a version
of the model covering a western European domain (35° N – 70° N latitude, 15° W - 35° E longitude). The whole
troposphere is described from ground to about 200 hPa using 30 hybrid (sigma, P) levels. The meteorological
forcing is given by the IFS forecast of the ECMWF based each day on the 0 and 12 am analysis. The
anthropogenic emissions are prescribed by using the TNO inventory (Kuenen et al, 2014) while natural
emissions are calculated by the MEGAN module (Guenther et al, 2006). A passive tracer has been used to
analyse the dynamics patterns responsible for ozone transport, which is initialized and emitted in the top model
layer (11-12 km) i.e within the upper troposphere every hour. A 10- day period (spin-up period) is simulated
before each targeted period to establish a kind of equilibrium state of the tracer. Moreover, simulations have also
been made by switching off emissions for both periods under study. Differences between simulations with and
without emissions is a proxy of the photochemical production of ozone within the boundary layer.
The methodology used in this paper is the following:
At first, in order to minimize local pollution effects and focus on boundary layer ozone measurements
representative of a wider geographical area than the station location, only the afternoon (12:00 – 18:00) ozone
concentrations, typically representing a well-mixed boundary layer, have been analyzed.
Also, recent observations over the Mediterranean both from MOZAIC tropospheric ozone profiles (Kalabokas et
al., 2013) as well as satellite data (Zanis et al., 2014) show a strong anticorrelation between ozone and
atmospheric humidity. Therefore, in dry and descending air masses originating from the upper tropospheric
layers, higher tropospheric ozone levels would be expected. It is well known that the concentration of water
vapour in the troposphere (specific humidity) tends to decrease with increasing altitude because the lower air
temperatures at higher altitudes cause elimination of water vapour by condensation followed by precipitation.
This makes specific humidity an indicator of subsiding air masses, which will be used in the analysis.
In addition, previous research carried out in the Mediterranean  (Kalabokas et al., 2007; Kalabokas et al., 2008;
Velchev et al., 2011; Kalabokas et al., 2013; Kalabokas et al., 2015) suggested that there is a strong link between
synoptic meteorology conditions and ozone concentration variability.
Based on the above, a systematic investigation of the composite meteorological maps (NCEP/NCAR Reanalysis)
during two spring high ozone episodes at the 850 hPa pressure level was carried out, until 5 days before the
event for the meteorological parameters mentioned previously. The purpose of the analysis is to identify areas of
high subsidence in the free troposphere, which could potentially influence the examined surface ozone
measurements, considering also that the afternoon (12:00 – 18:00) ozone concentrations are quite representative
of the boundary layer values. The high subsidence areas were detected in the first place by the positive vertical
velocity omega (and anomalies) as well as the negative specific humidity anomalies. Also, the geographical
distributions of geopotential heights, temperature anomalies and vector wind speed were very useful in the
examination of the influence of synoptic meteorological conditions on ozone concentrations. In addition, air
mass back trajectories (NOAA HYSPLIT), satellite IASI ozone measurements at the 3km and 10km levels,
afternoon surface ozone measurements from the EEA European network and vertical ozone profile
measurements in Central Europe during a selected ozone episode, were used for the analysis. Finally, CHIMERE





tracer simulations and modelling of ozone field for the April and May 2008 ozone episodes have been performed
for the validation of the analysis of measurement data.
**3  Results and discussion**
**3.1  Surface Ozone measurements in the Western Mediterranean basin in spring 2008**
In Fig. 1 (upper panel) the afternoon (12:00 – 18:00) rural ozone concentrations of selected EMEP stations from
countries surrounding the Western Mediterranean basin (Spain, France, Switzerland, Italy, Malta) during spring
(March – May) 2008 have been plotted. A first investigation of the plots leads to the following remarks:
a). Episodic periods of high ozone (and also for low ozone) may last for several days and they can be detected
simultaneously in many countries surrounding the western Mediterranean basin.
b). The 60 ppb (120 $\mu g/m^3$) EU standard for human health protection can be exceeded over many days and in
many countries.
From the above stations, the ozone concentrations measured at the EMEP stations in France and Switzerland as
well as the EMEP station in Northern Italy (IT04, JRC-Ispra), presented in Fig. 1 (lower panel), show generally a
very good agreement between each other. This feature is remarkable given their different site characteristics, the
large distance between them as well as their location relative to the huge natural barrier of the Alpine
mountainous area. So, the high mid-day ozone concentrations observed simultaneously over many countries
could be considered as regional ozone episodes. The two regional spring 2008 ozone episodes with the highest
ozone concentrations in the examined area were on 26-27 April, 2008 and 7-9 May, 2008; they have been
selected for further analysis.
**3.2  Geographical distribution of meteorological parameters, IASI  tropospheric ozone measurements and**
**CHIMERE modelling tracer simulations during the April 26-27, 2008 ozone episode**
The composite NCEP/NCAR Reanalysis maps at 850 hPa for the high ozone episode of April 26-27, 2008 (and
also 2, 3 and 5 days before) are presented in Fig. 2 (geopotential height/specific humidity anomaly), Fig. 3
(vertical velocity omega/vertical velocity omega anomaly) and Fig. 4 (temperature anomaly/vector wind speed).
The corresponding composite maps at the 700hPa and 500 hPa pressure levels have also been plotted and
analyzed (not shown). The examination of these meteorological charts leads to the following remarks:
a). The anticyclone located to the west of the N. African coast is progressively strengthening and expanding after
April 23-24, 2008 while moving rapidly towards the European continent. At the same time the low pressure
system installed over E. Europe on April 21-24 is progressively reduced to a cut-off low on April 26-27 while
moving towards Greece and Turkey (Fig. 2).



b). Negative specific humidity anomalies are observed all over N. and E. Europe as well as over the Atlantic
Ocean while at the peak of the episode (April 26-27) the dry air masses prevail over a vast and extended area
from Russia to NW Africa and the Atlantic (Fig. 2).
c). An extended area of positive vertical velocity omega (and anomalies), which is a direct sign of subsiding air
masses, is observed over the W. Mediterranean and the Atlantic and is moving rapidly towards the Central
Mediterranean. Five days before the episode (April 21-22) the maximum of subsidence area was observed over
the Iberian Peninsula and the adjacent ocean up to the Canary islands. Then it is moving rapidly eastwards,
showing a maximum over Italy at the peak of the episode (April 26-27) as well as a secondary maximum over
Germany (Fig. 3). The positive omega anomalies indicate that the observed subsidence during the episode days
is higher than usual for this period of the year and the examined geographical regions.
d). A strong westerly flow is observed over the Western Mediterranean region of subsidence with the air masses
originating from the region of N. Atlantic where a deep and extended low pressure system is observed (Fig. 4).
As seen, the extension of the N. African anticyclone towards the Western Mediterranean changes progressively
the very strong westerly flow (observed 5 days before the episode, April 21-22) to a strong north-westerly. Also,
strong N-NW winds over the Central Mediterranean prevail at the interface area between the cut-off low and the
anticyclone.
e). Strong and extended negative temperature anomalies are observed 5 days before the episode over the Iberian
peninsula and the adjacent N. African coast, moving towards the Central Mediterranean and the Libyan coast.
The negative temperature anomaly area becomes more extended during the episode days, indicating that colder
atmospheric conditions than the normal prevail during the ozone episode over the corresponding areas, while
during photochemical ozone episodes higher than normal air temperatures would be expected. This can be
explained as an impact of transport of cold air masses to these areas.
Overall, over the same area of subsidence, strong winds together with positive omega vertical velocity (and
anomalies) are observed as well as negative humidity anomalies, thus indicating a strong descending air current,
which transports rapidly tropospheric air towards the boundary layer.
In Figure 5, the composite ozone IASI measurements from April 21, 2008 to April 27, 2008 are presented at two
altitude levels (3 km and 10 km), which are considered representative for the lower and the upper troposphere
(Doche et al., 2014) ), even if a partial overlap occurs due to reduced resolution (see section 2). An extended area
of very high ozone in the upper and lower troposphere is observed over the N. Atlantic over the area covered by
the low pressure system. At the same time, in the Central Mediterranean a high tropospheric ozone area is
progressively formed at both examined tropospheric layers (3 and 10 km) and having its maximum on April 26-
27. This observation could be associated with the information extracted from the analysis of the composite
meteorological charts over this Central Mediterranean area where clear indications of very strong subsidence of
dry air masses is observed together with very strong north-westerly winds recorded over the Mediterranean at
850hPa (Figs 2-4), especially for April 26-27. The same features are also observed at the pressure levels of
700hPa and 500 hPa (not shown).
For a more detailed analysis during the maximum of the episode (April 26-27), in Fig. 6 the daily IASI satellite
ozone measurements at 3 and 10 km altitude are presented in a higher concentration resolution as well as in a





more restricted geographical domain (same as the CHIMERE simulation outputs), where the above described
characteristics of the geographical distribution of tropospheric ozone appear more clear. Also, for a more
detailed investigation of the corresponding meteorological characteristics at higher tropospheric levels, in Fig. 7
the daily meteorological charts of columnar precipitable water anomaly and geopotential heights at 500hPa and
700hPa for April 26 and 27, 2008 are presented. The very extended areas with negative columnar precipitable
water anomalies show that the dry conditions, indicating subsidence, prevail over the whole troposphere, mostly
over its lower part, over a vast area and especially at the interface areas of the high and low pressure systems at
500 hPa, very similar to what was already observed at 850 hPa (Fig. 2). So, it comes out that during the ozone
episode the same synoptic pattern is observed in the troposphere up to 500 hPa (Fig. 7) but even at higher levels
(not shown).
In order to study in more detail the atmospheric processes prevailing during the examined April ozone episode
and complete the analysis of the dynamic atmospheric conditions during the examined events, regarding more
specifically the origin of ozone measured in the boundary layer, we have performed simulations with the
CHIMERE model using tracers to analyze transport patterns. The results of the CHIMERE simulations for the
April episode are shown in Figs 8-9, where simulations of upper tropospheric tracer concentrations, simulations
of photochemical production and simulations of the ozone field (together with the iso-contours of the high
tropospheric tracer) are presented. As described earlier (cf Chapter 2), we use a tracer initialized within the
model top layer in the upper troposphere at about 11 km height. Inspecting qualitatively simulated
concentrations at 3 km and 1.5 km altitude (Fig. 8, 9), it is shown that the upper tropospheric tracer is present at
these levels, indicating significant downward transport or upper troposphere air masses to the boundary layer,
which coincides with the maximum concentrations of the CHIMERE simulated ozone field at 3 and 1.5 km
covering a large area from Switzerland to Malta (Fig. 9). Combining that with the meteorological charts (Figs 2,
7), on the outer side of the low-pressure area and at the interface with the N. African anticyclone high values of
the upper tropospheric tracer indicating considerable subsidence are observed at 5 km altitude (not shown),
which becomes even stronger and more extended at the 3 km and 1.5 km altitudes (Fig.9). So, following the
geographical distribution of the upper tropospheric tracer at 3 km and 1.5 km (Fig.9), tropospheric subsidence is
observed at the periphery of the anticyclonic area and at the interface with the low-pressure area. Therefore, the
downward transport of upper tropospheric ozone is influencing the boundary layer over the examined location.
The observed patterns, based on CHIMERE tracer simulations and on ozone concentrations are quite consistent
with the observed values of the IASI instrument that show the same structures, in particular enhanced
concentrations at 3km over Italy (Figs 5,6), as well as with the analysis of meteorological parameters based on
composite charts (Figs 2-4) and described above. Regarding the comparison of the geographical distribution of
the CHIMERE ozone field (Fig.9) with the IASI measurements at 3 km or 700hPa (Figs 5-6), a clear difference
over the N. Atlantic and over the Central Mediterranean is observed. As discussed above, in both regions the
prevailing meteorological conditions (low pressure systems) are associated with high tropospheric ozone levels.
It has to be mentioned that direct comparison of CHIMERE simulations at 3 km with the corresponding IASI
measurements presents some weaknesses, especially due to the fact that IASI is sensitive to a height range
between 2 and 8 km and the IASI averaging kernels need to be applied to CHIMERE to make it comparable with
IASI (Eremenko et al., 2008).





The graphical representation on the map of the daily evolution of the hourly surface ozone concentrations at
15:00 h as recorded in the air pollution stations of the EEA-AirBase network (EEA, 2015) during the 26-27
April period (Fig. 9) shows very high concentrations, exceeding 70 ppb, appearing at many locations in Spain as
well as in S. France and NW Italy. As mentioned previously, during the afternoon hours the influence of the free
troposphere on the boundary layer is maximized through vertical mixing, and the composite meteorological
maps show a large corridor of subsiding dry air masses over the same area (Figs 2-4), which is a very strong
indication that the surface ozone concentrations could be influenced by entrainment of subsiding ozone-rich air.
The CHIMERE tracer simulations suggest that in this case the subsiding air masses are not coming directly from
the upper troposphere.
The HYSPLIT back-trajectories during the April episode arriving at the EMEP stations in Italy (JRC-Ispra) and
on Malta (Fig. 10) show in fact subsiding air masses arriving from northern directions, on April 25 and 26 for
Ispra and on April 27 for Malta and give more or less the same picture as the Swiss stations (not shown).
The combination of the information from the composite meteorological maps (Figs 2-4) and the HYSPLIT back-
trajectories for the April episode (Fig. 10) shows that the prevailing NW wind during the days preceding the
episode passed over the N. Atlantic region, where the IASI satellite detects a high ozone area extended
throughout the whole troposphere (Figs 5, 6), which beyond any doubt contributes to the high surface ozone
values.  So, tropospheric transport of air rich in ozone from northern directions is likely to give an important
contribution to the high surface ozone levels through the processes of advection and subsidence.
It should be noticed at this point that the interpretation of the back trajectory information is much more efficient
if it is done in combination with the examination of the composite meteorological charts (and ideally satellite
ozone measurements and modelling simulations). In this way the influence from areas with strong tropospheric
subsidence and low atmospheric humidity (indicating high ozone) at the various tropospheric pressure levels
could be easier assessed.
In summary, the April 26-27, 2008 ozone episode could be considered as a clear case of the tropospheric
influence to the boundary layer through ozone transport, which is added to the photochemically produced
boundary layer ozone. The analysis of meteorological charts at various tropospheric pressure levels helps in
understanding that the strong subsidence occurring over the central Mediterranean plays a key role in explaining
this surface ozone episode on April 26-27, 2008. The IASI observations of lower and upper tropospheric ozone,
helps in understanding that the subsidence of ozone-rich air masses characterizes this surface ozone episode. The
evolution of the above described phenomenon can be effectively monitored by examining the composite
meteorological maps at 850hPa during the episode days (as well as at 5-days, 3-days and 2-days before the
episode). As mentioned, the examination of the 700hPa and 500hPa pressure levels (not shown) gives
comparable results.
Based on the above results, the simultaneous strengthening and expansion of the African anticyclone and the
formation of a cut-off low over the SE Balkans during the episode seem to give rise to the large scale subsidence,
inducing a strong downward transport of cold air masses over the western Mediterranean and the surrounding
areas while the maximum intensity of the phenomenon is observed in the geographical area located between the
two synoptic atmospheric systems. As mentioned, the signs of stronger tropospheric subsidence during the





episode days can be clearly observed from the negative specific humidity anomalies as well as the negative
columnar precipitable water anomalies associated with positive omega anomalies (downward motion). The
evolution of the temperature anomaly is interesting as the growing region of negative temperature anomaly over
more or less the region of subsidence (mostly to its west) as the episode develops, could be explained by the
descend of much colder air from the upper tropospheric layers caused by the interaction of the low pressure
system (located to the east) and the anticyclone (located to the west). This is also consistent with the descent of
upper tropospheric tracers simulated with the CHIMERE model. All the above features accompanying a situation
with deep subsidence appear in all examined pressure levels at 850 hPa, 700hPa and 500 hPa.
**3.3  Geographical distribution of meteorological parameters, IASI tropospheric ozone measurements and**
**CHIMERE modelling tracer simulations during the May 7-9, 2008 ozone episode**
As it will be shown in the next paragraphs, the examination of the May 7-9, 2008 episode leads to comparable
remarks regarding synoptic meteorology and geographical distribution of meteorological parameters with the
April episode, although in the May episode the meteorological systems and especially the anticyclone are located
further to the north.  In Figures 11, 12 and 13 the composite NCEP/NCAR Reanalysis maps for the high ozone
episode of May 7-9, 2008 at 850 hPa as well as for 2, 3 and 5 days before are presented while the same plots for
the 700 hPa,  500 hPa and 300 hPa level have been also created (not shown). Their examination gives the
following:
a). A deep and extended low is located over N. Atlantic while a strong anticyclone prevails over N. Europe,
centered between Great Britain and Scandinavia. At the same time deep and extended low-pressure systems over
the polar regions and E. Europe are formed, associated with very dry conditions observed throughout the whole
troposphere (Fig. 11).
b). Over a part of the anticyclonic area and especially at the region of interface of the anticyclone with the low
pressure systems to the east (Fig. 12), an extended area of positive vertical velocity omega (and anomalies) is
observed, associated also with extended areas of low-humidity air masses (Fig. 11), which are clear signs of
strong subsidence lasting for many days. The signs of subsidence are particularly strong also at the higher
tropospheric pressure levels up to 300hPa (not shown) indicating the occurrence of a large-scale tropospheric
phenomenon.
c). A strong contrast is observed (Fig. 13) between positive and negative temperature anomalies in W. and E.
Europe respectively (and also geopotential height anomalies, not shown), which is similar to what has been
observed for the April episode, showing strong influence of vertical transport of air masses which tends to occur
at the interface areas between high and low atmospheric pressures systems as well as between the associated
positive and negative temperature anomalies.
d). Stagnant conditions are observed at the center of the N. European anticyclone located over the North Sea
while at the southern part of the anticyclone a strong positive vertical omega velocity is observed (indicating
subsidence), which is associated with a stronger vector wind velocity over C. Europe. In fact, a strong air current





starting from the N. Atlantic makes a circular clockwise motion at the periphery of the anticyclone towards the
East while the flow at the northern and the western periphery of the anticyclone is particularly strengthened (Fig.
13). Similar features are also observed at the higher pressure levels (700hPa, 500 hPa and 300hPa) as it was also
the case for the April episode.
In Figure 14, the IASI measurements (composite charts) during the May 7-9, 2008 episode for the lower and the
upper troposphere (3 and 10km respectively) as well as for the previous 2, 3 and 5 days are shown. During the
May episode a high ozone area is observed in the lower and upper troposphere over the N. Atlantic, which is
more extended during the days preceding the episode peak.
As observed in Figure 14, the days before the episode and over the region covered by the anticyclone, the
concentrations of ozone in the upper troposphere (at 10 km) decrease, while the IASI measurements in the lower
troposphere (at 3 km) over central Europe show a small but geographically extended increase. It has to be
reminded that this phenomenon takes place exactly over the area where intense subsidence and low humidity
conditions have been detected, which could be considered as a direct independent evidence of the influence of
ozone concentrations from the upper troposphere, as it has been described and discussed in the previous
paragraphs. As also seen in Figure 14, over E. Europe and Russia a significant tropospheric ozone accumulation
occurs while a prevailing easterly flow from these high ozone areas is moving towards central and W. Europe
(Fig. 13). It should be reminded that over central Europe extended strong subsidence associated with dry
conditions and a strong gradient of temperature anomalies are observed at the same time (Figs 11-13).
The above features are more clearly shown in Fig. 15 where the daily IASI satellite ozone measurements at 3 and
10 km level for the May 2008 episode are presented in a higher concentration resolution as well as in a more
restricted geographical domain (same as the CHIMERE model outputs) while the corresponding daily
meteorological charts of columnar precipitable water anomaly and geopotential heights at 500hPa and 700hPa
are shown in Fig. 16. In fact, an extended region of negative columnar precipitable water anomalies (dry
conditions throughout the troposphere, indicating subsidence), prevailing over a vast area covering eastern and
central Europe is observed. As also seen in the April episode, this feature appears at the interface areas of the
high and low pressure systems while the same synoptic patterns occur at all examined tropospheric levels.
The results of the CHIMERE simulations for the May episode are shown in Fig. 17-18, where simulations of
upper tropospheric tracer concentrations, simulations of photochemical production and simulations of the ozone
field (together with the iso-contours of the high tropospheric tracer) are presented. As described earlier (cf
Chapter 2), we use one tracer initialized within the model top layer in the upper troposphere at about 11 km
height. Inspecting qualitatively simulated concentrations at 3 km and 1.5 km altitude (Fig. 17, 18), it appears that
the upper tropospheric tracer is clearly detected at these levels, indicating downward transport from the upper
troposphere to the boundary layer. Also in Fig. 18, the CHIMERE simulations of the ozone fields at 3 km and
1.5 km altitudes (corresponding to 700hPa and 850hPa pressure levels respectively) show high ozone
concentrations over C. Europe, Italy and the W. Mediterranean. More precisely, the highest upper tropospheric
tracer influence occurs to the south of the N. European anticyclone and it is progressively strengthened between
7 and 9 of May while high ozone is observed over the whole anticyclonic area. In general, over that area the
observed high values of the upper tropospheric tracer indicate considerable subsidence at 5 km altitude (not





shown), which becomes even stronger and more extended at the at 3 km and 1.5 km altitudes (Fig. 18).
Therefore, the downward transport of upper tropospheric ozone is influencing the lower troposphere and the
boundary layer over the examined location. The observed patterns, based on CHIMERE tracer simulations are
quite consistent with the above described analysis based on meteorological charts (Figs 11-13) and IASI satellite
measurements (Figs 14, 15).
Regarding the comparison of the CHIMERE ozone field at 3 km with IASI at 700hPa there is in general a good
agreement over Italy and the central Mediterranean but there are differences in E. Europe (lower CHIMERE
values) and N. Europe (higher CHIMERE values), which might imply an underestimation of tropospheric
transport in E. Europe or overestimation of photochemistry in N. Europe. It should be reminded that similar
features appear also in the April episode, described previously. Nevertheless, as mentioned also for the April
episode, it should be kept in mind that for a direct quantitative comparison between CHIMERE and IASI data,
an application of IASI averaging kernels (smoothing functions) to CHIMERE output would be required
(Eremenko et al., 2008).
The graphical representation on the map of the daily evolution of the hourly surface ozone concentrations at
15:00 h as recorded in the air pollution stations of the EEA-AirBase network (EEA, 2015) during the 7-9 May
period (Fig. 18) shows very high concentrations, exceeding 70 ppb, appearing simultaneously at many locations
from N. Italy to the British islands. This is in a very good agreement with the CHIMERE tracer simulations as
well as the composite meteorological maps (Figs 11-13, 16) and the IASI satellite measurements (Figs 14-15),
showing an extended region of subsiding dry air masses over eastern and central Europe. As mentioned
previously, during the afternoon hours the influence of the free troposphere on the boundary layer is maximized
through vertical mixing and thus the surface ozone concentrations could be increased by the entrainment of
subsiding ozone-rich air.
In Figure 19 the HYSPLIT back-trajectories during the episode at EMEP stations in Italy (IT04) and France
(FR10, FR14) show subsiding air masses arriving either from the north after performing a circular clockwise
movement or  from the NE (E. Europe and Russia) where high tropospheric ozone concentrations have been
recorded by IASI (Figs. 14-15). This is an additional confirmation that air masses originating from areas with
high tropospheric ozone concentrations transport ozone down to the ground, leading to high surface ozone
values.
**3.3.1 Ozone vertical profiles over Payerne, Uccle and Hohenpeissenberg during the May 2008 ozone**
**episode**
Due to the more complex nature of the May 7-9, 2008 ozone episode, the vertical ozone profiles at three
European ozone sounding stations (Payerne-Switzerland, Hohenpeissenberg-Germany and Uccle-Belgium) were
taken into account for the analysis and the respective vertical ozone measurements for each station are shown in
Fig 20.
It has to be reminded that the Payerne site is one of the two Swiss stations, where the rural afternoon surface
ozone values during the May episode were about 75 ppb (Figs 1-2). The composite meteorological maps (Figs



11-13) indicate that the Payerne site was within the area influenced by subsidence. As observed in Fig. 20 a
layer with the tropospheric ozone maximum concentration and low relative humidity is at the beginning of the
event (on May 5) located between 5 and 6 km altitude. On May 7 a similar layer is seen at a somewhat lower
altitude and on May 9 the tropospheric ozone maximum is found below 2000 m altitude. It has to be added that
on May 12, at the end of the episode, the vertical ozone profile has changed completely and the ozone
concentrations up to 6 km were about 60 ppb (not shown). As mentioned in the meteorological analysis of the
episode (Figs 11-13), the downward ozone transport from the ozone-rich lower troposphere to the boundary
layer during the 5-9 May period with strong and persistent subsidence, is reflected clearly in the ozone profiles
over Payerne (Fig. 18). Based on the observed conditions the plausible explanation is that an "ozone fumigation"
of the boundary layer occurred between 5-9 May when the ozone levels increased by at least 20 ppb, following
corresponding changes in the lower troposphere and indicating that this ozone event is related to downward
transport of ozone from higher altitudes towards the boundary layer.
Similar observations could be made at Hohenpeissenberg and Uccle during the May 5-9 period (Fig. 20) where
high ozone layers are also observed in the lower troposphere (around 80 – 100 ppb at 3-5 km on May 5 over both
sites) and which clearly move downwards. The observed pattern is in agreement with the strong and persistent
subsidence observed over the area during the examined days, as it was the case for Payerne.   As shown in the
vertical ozone soundings (Fig. 20) the high ozone tongue in the troposphere is going down rapidly at a 1-2 km
/day rate. At the same time over the examined area the positive vertical velocity omega (and anomalies)
observed are associated with dry air (Figs 11-13), indicating subsidence from the upper troposphere all the way
down to the surface.
It has to be noticed at this point that vertical ozone profiles over the airports of Frankfurt and London carried out
in the framework of the MOZAIC project (Marenco et al., 1998; Thouret et al., 2006) during the examined
period show almost exactly the same picture (not shown).
Taking into account all the above information regarding the May episode, the following remarks regarding the
tropospheric ozone distribution could be made:
a). The IASI satellite measurements shows that over N. Atlantic and to the west of the N. European anticyclone
(present at all tropospheric pressure levels up to 300hPa) there are high amounts of ozone in the upper and lower
troposphere.
b). The anticyclone transports these air masses from the west to the east in a clockwise rotating movement.
c). Over E. Europe and at the southern part of the anticyclone the air masses show strong signs of subsidence
(downward vertical velocities, dry air), which is particularly enhanced in the vicinity of the strong low-pressure
system over eastern Europe.
d). A significant ozone decrease in the upper troposphere is observed over the region covered by the N.
European anticyclone followed by a slight but extended ozone increase in the lower troposphere, located over the
area with intense subsidence (Central Europe) to the south of the anticyclone, with a several day time-shift.





In summarizing the observations on the May 7-9, 2008 episode an important question is to which extent the high
surface ozone levels observed at first in countries surrounding the western Mediterranean basin but also over a
large area in central Europe are influenced by entrainment of ozone rich air from higher layers of the
troposphere. An interesting feature is that during the examined period an extended area of strong and persistent
downward movement of air masses is observed over Central Europe and lasts for many days. Another important
observation is that the wind makes a circular downward motion around the anticyclone while this air current is
influenced from regions of very high tropospheric ozone surrounding the anticyclone (located over the observed
low pressure systems), according to IASI observations. The high ozone areas coincide very well with extended
areas of low humidity as observed in the 850hPa level charts but also at higher tropospheric levels. At the same
time, the vertical ozone profiles over Payerne, Uccle and Hohenpeissenberg show concentrations at 60-80 ppb on
the top of the boundary layer. It should be noticed that all the above sites are located more or less within the
region of high subsidence as the meteorological analysis shows. The CHIMERE tracer simulation experiments
are in very good agreement with the above observational analysis. As observed, during this episode the high
measured tropospheric ozone background values contribute very significantly, through subsidence, to enhanced
ozone values especially over France, Switzerland, northern Italy, and the Western Mediterranean basin, while
regional ozone photochemical production is the dominant factor for ozone enhancement over the British Islands,
the North Sea as well as on the Iberia peninsula and parts of France (Fig. 17).
Summarizing all the above, during the examined May episode as well as in the previous April episode, high
atmospheric pressures prevail over the Western Mediterranean and Central Europe while low-pressure systems
are observed in Eastern Europe. In general, both examined episodes (26-27 April 2008 and 7-9 May 2008) seem
to show comparable characteristics: At first, very strong positive omega anomalies (strong downward air motion
or subsidence) are observed over large areas including the Italian peninsula and Central Europe. At the same
time, strong negative specific humidity anomalies and negative precipitable water anomalies (indicating dry air
in the troposphere) are detected. An interesting feature is that the subsidence area is located at the interface of
positive and negative anomalies of geopotential heights and temperature. For the May episode, although the
major anticyclone seems to be located away from the Mediterranean basin to the north, a large-scale downward
transport towards the Mediterranean according to the meteorological analysis is observed. It comes out from the
above analysis that the interpretation of the back trajectory information is much more efficient if it is done in
combination with the examination of the meteorological charts (and ideally supported by satellite measurements
and modelling simulations), so that the regions with strong subsidence and dry air, are detected and taken into
account in the interpretation.
**3.4 Diurnal variation of surface ozone and humidity during the ozone episodes**
The measurements of surface ozone, relative humidity and temperature at the EMEP site at Ispra, Italy, (Jensen,
2016) have been used during the April and May episodes in order to observe the relationship between ozone and
absolute humidity (Fig. 21), which could be a good indicator for the detection of the free tropospheric influence
to the boundary layer being insensitive to the diurnal temperature variation. Absolute humidity is shown in units
of hPa water vapor partial pressure, calculated from the relative humidity using the August-Roche-Magnus



formula to find the saturated water vapor partial pressure at the measured temperature. The ozone concentrations
show a characteristic diurnal variation with a minimum during the night, followed by a rapid increase in the
morning when the nocturnal boundary layer breaks down and air from the above layer reaches the ground. For
all days with high ozone during the May episode and for some days during the April episode the ozone increase
at mid-day with the maximum vertical mixing is clearly anticorrelated with absolute humidity. This is
compatible with the fact that the Ispra site, according to the meteorological analysis presented above, during the
whole duration of the May episode is expected to be influenced by subsidence of dry air while during the April
episode much stronger anticorrelations and lower absolute humidity levels are observed for some days. Based on
the presented example, a first identification of possible direct tropospheric influence to the boundary layer air
could be performed at air pollution monitoring stations potentially influenced by tropospheric subsidence.
**4 Conclusions**
In this paper, the investigation of the regional ozone episodes in the western Mediterranean and the surrounding
countries is focused on the spring season.
Based on the ozone measurements at the EMEP rural ozone stations surrounding the western Mediterranean
basin (as well as the EEA-AirBase network), it is observed that episodic periods of high ozone may last for
several days and they can be detected at several countries at the same time.
An interesting result was that the examined ozone episodes are linked to meteorological conditions very similar
to those observed in the eastern Mediterranean (Kalabokas et al., 2013; Doche at al., 2014; Kalabokas et al.,
2015), related essentially with transport of ozone-rich tropospheric air masses and the atmospheric subsidence
phenomenon. Based on the analysis of the selected springtime ozone episodes, the characteristics associated with
high ozone concentrations close to the surface through important subsidence and in connection with high
tropospheric ozone levels occurring in the wider area, are the following:
The geographic areas with observed deep tropospheric subsidence seem to be the transition regions between a
high pressure system, located in the west sector, and a low pressure system located in the east sector, as shown in
the corresponding charts of the geopotential heights. Over these areas, strong gradients of geopotential height
and temperature are observed together with high omega vertical velocity values and low specific humidity values
at the 850hPa as well as at higher tropospheric pressure levels. In addition, over the areas of deep tropospheric
subsidence, negative temperature anomalies are observed at these levels and also high vector wind speeds, which
means that subsidence is associated with strong advection. The above observational analysis is in very good
agreement with IASI satellite measurements and CHIMERE tracer simulation experiments.
The present approach, using meteorological charts, IASI tropospheric ozone satellite measurements, CHIMERE
tracer simulations and back trajectories for the analysis of selected spring ozone episodes, shows to be quite
efficient in the analysis of atmospheric conditions and transport patterns associated with the episodes, which
need to be adequately described in numerical chemical-transport models used for simulations of air pollution. It
can be useful for the study of the tropospheric influence on the boundary layer and the ground surface, especially
for tracing large scale deep subsidence events (downward movements of generally dry air masses) when



analyzing surface (and vertical, if available) measurements at sites in the Mediterranean basin, where this
phenomenon is quite frequent, and also in other places worldwide. For example, similar field observations on the
influence of upper atmospheric layers on the boundary layer and surface measurements have been reported at
some locations in the United States (Parrish et al., 2010; Cooper et al., 2011; Langford et al., 2015). In addition,
the consideration for the analysis of ozone measurements presented above is in agreement with recent
observations in Atlantic and European regions indicating significant tropospheric (and even stratospheric)
influence to the surface and boundary layer ozone measurements (Trickl et al., 2010; Trickl et al., 2011; Logan
et al., 2012; Cuevas et al., 2013; Hess and Zbinden, 2013; Cooper et al., 2014; Parrish et al., 2014).
Regarding the environmental policy issues, it has to be underlined that for ozone, which is a pollutant regulated
by the EU, it appears that there are some time periods during the warm period of the year, lasting for several
days, when the free troposphere influences significantly the boundary layer to such extent that the air quality
standards might be exceeded. This phenomenon seems to be associated with an important impact of
photochemical ozone production following primary air pollutant emission at larger geographical scales,
including transport of air pollutants on a hemispheric scale from e.g. the US and China. The origin of the
atmospheric ozone entering the boundary layer might be upper tropospheric or stratospheric but it could also be
from the lower or the middle troposphere during stagnant regional conditions when photochemical ozone is
produced over the continent.
Further detailed and quantitative studies of this phenomenon appear to be needed to improve our understanding
of the mechanisms associated with intercontinental pollution as well as regional photochemical pollution could
be improved, already highlighted by the work of the HTAP (Hemispheric Transport of Air Pollution) project.
The Mediterranean region which seems to be mostly affected is the Eastern Mediterranean, where the 60 ppb EU
standard is very frequently exceeded during the warm period of the year and especially during July-August when
the atmospheric subsidence is a quite common atmospheric feature and almost quasi-permanent, but as shown in
the present study, also in the Western Mediterranean a similar mechanism could be observed, especially during
the spring season. The strategies for obtaining compliance with the EU ozone standard may need to be
reconsidered by taking into account the contribution from regional and intercontinental transport.
**Acknowledgements**
The composite weather maps were provided by the NOAA/ESRL Physical Sciences Division, Boulder Colorado
from their Web site at http://www.cdc.noaa.gov/., which is gratefully acknowledged. The authors also
acknowledge the NOAA Air Resources Laboratory (ARL) for the provision of the HYSPLIT transport and
dispersion model and/or READY website (http://www.ready.noaa.gov) used in this publication. Also, Dr. A.
Volz-Thomas, FZ-Juelich, is acknowledged for interesting discussion as well as Dr. C. Repapis, Academy of
Athens, for useful comments on the manuscript. JRC is acknowledged for an EU grant (cat. 40) to one of the
authors (P. K.).



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



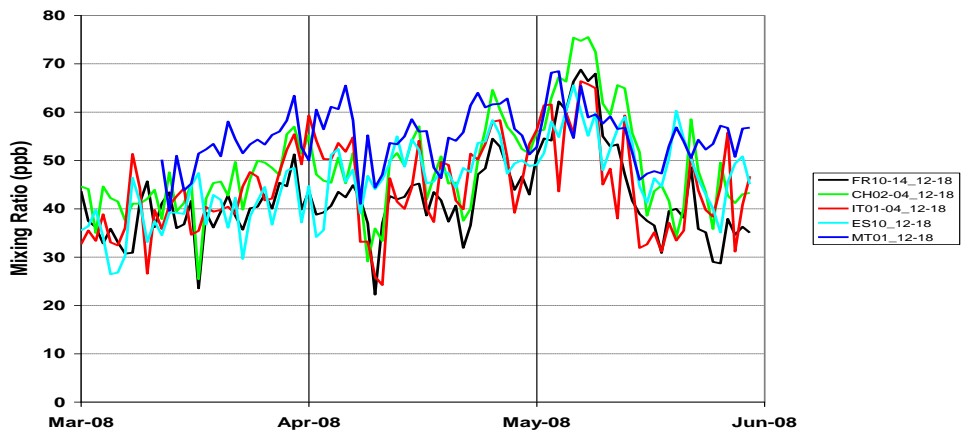

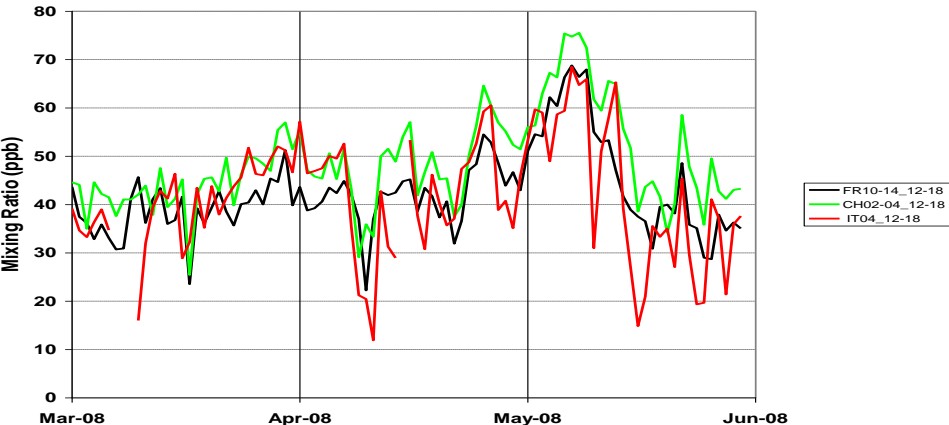

**Figure 1: (Upper panel): Day-to-day variation of the 12:00 - 18:00 afternoon ozone (in µg/m3) on the**
**EMEP stations surrounding the Western Mediterranean basin for spring 2008: Spain (ES10, light blue),**
**France (FR10-14, black), Switzerland (CH02-04, green), Italy (IT01-04, red), Malta (MT01, blue).**

**(Lower panel): Day-to-day variation of the 12:00 - 18:00 afternoon ozone (in µg/m3) on the EMEP stations**
**close to the Western Mediterranean basin and around the Alpine region for spring 2008: France (FR10-**
**14, black), Switzerland (CH02-04, green), Italy (IT04-JRC, red).**





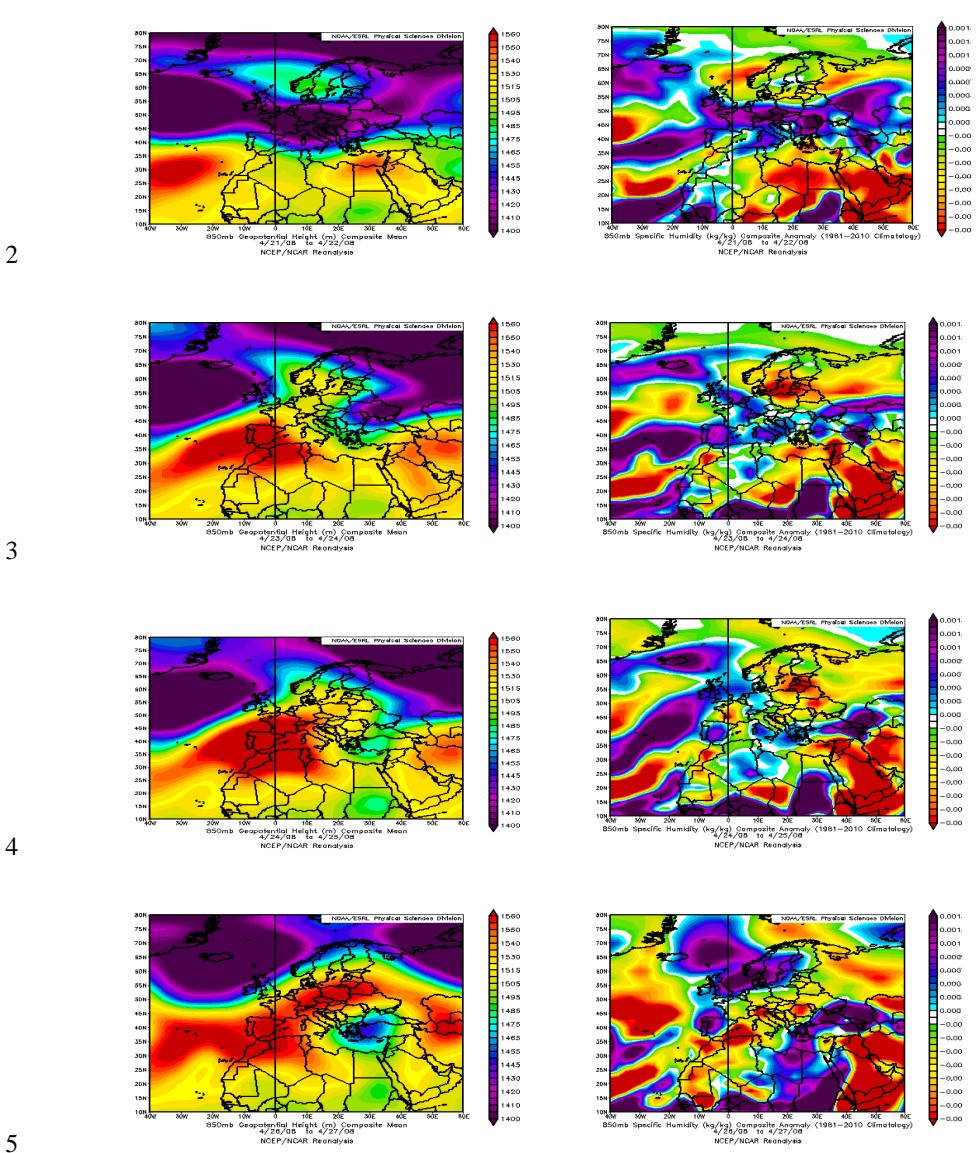

6   **Figure 2: Composite weather maps of geopotential height (left column) and specific humidity (right**

7   **column), for the high ozone episode of 26-27 April 2008 (lowest panels) as well as for two, three and five**

8   **days before.**





5 **Figure 3: Same as Fig. 2 but for omega vertical velocity (left column) and omega vertical velocity anomaly**

6 **(right column).**





5    **Figure 4: Same as Fig. 2 but for vector wind (left column) and air temperature anomaly (right column).**



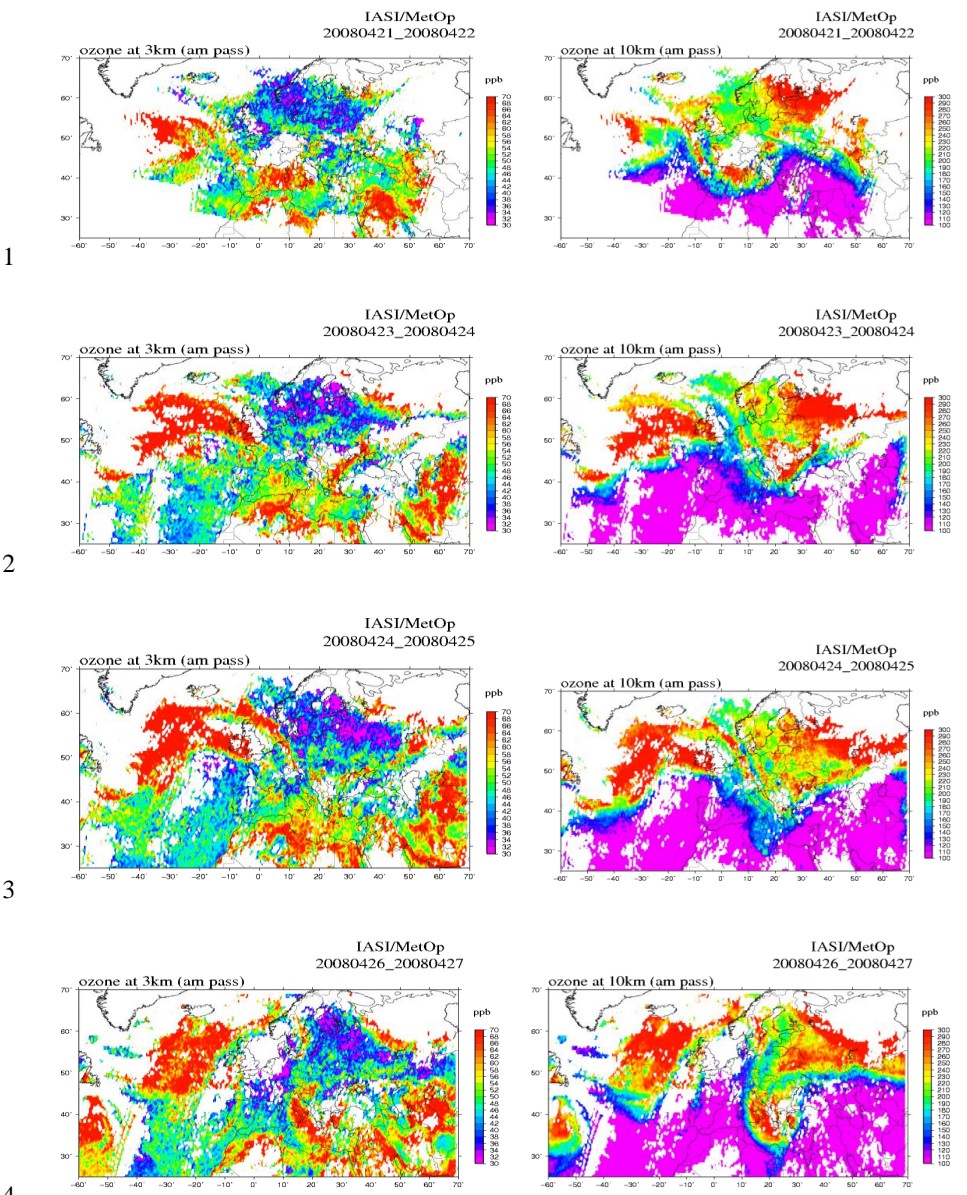

**Figure 5: IASI satellite ozone measurements at 3km level (left column) and 10 km level (right column) during the high ozone episode of 26-27 April 2008 (lowest panels) as well as for two, three and five days before. Values outside the scale range are set up to the upper and lower color code respectively.**





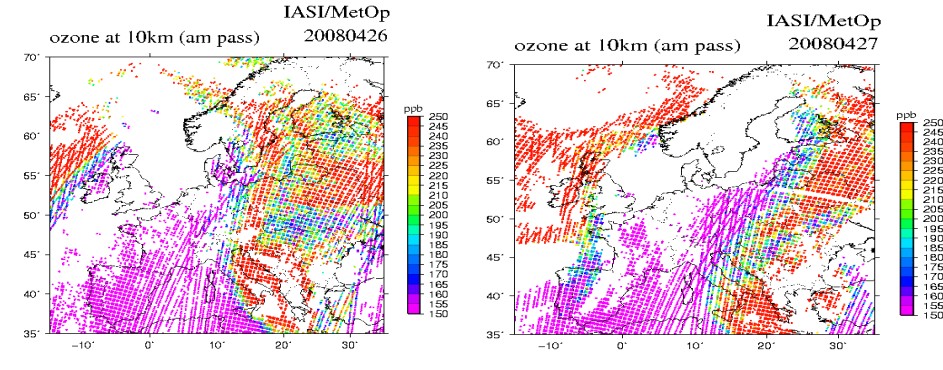

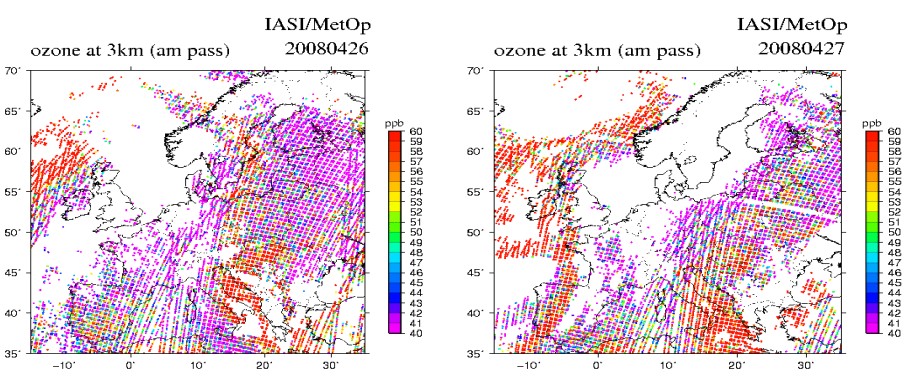

3  **Figure 6: Daily IASI satellite ozone measurements at 10 km level (upper panel) and at 3km level (lower**

4  **panel) for April 26, 2008 (left column) and April 27, 2008 (right column).  Values outside the scale range**

5  **are set up to the upper and lower color code respectively.**





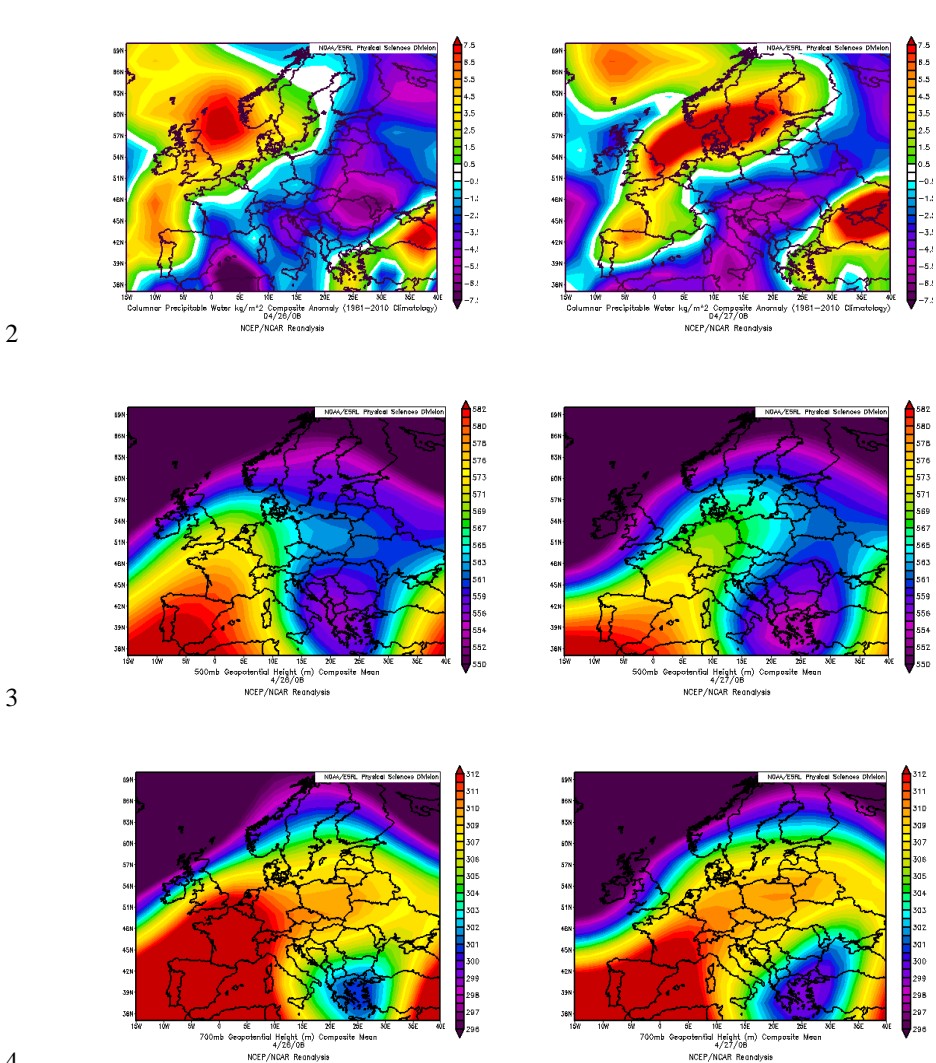

5  **Figure 7: Columnar precipitable water anomaly (upper panel), geopotential height at 500hPa (middle**

6  **panel) and geopotential height at 700hPa (lower panel) for April 26, 2008 (left column) and April 27, 2008**

7  **(right column).**



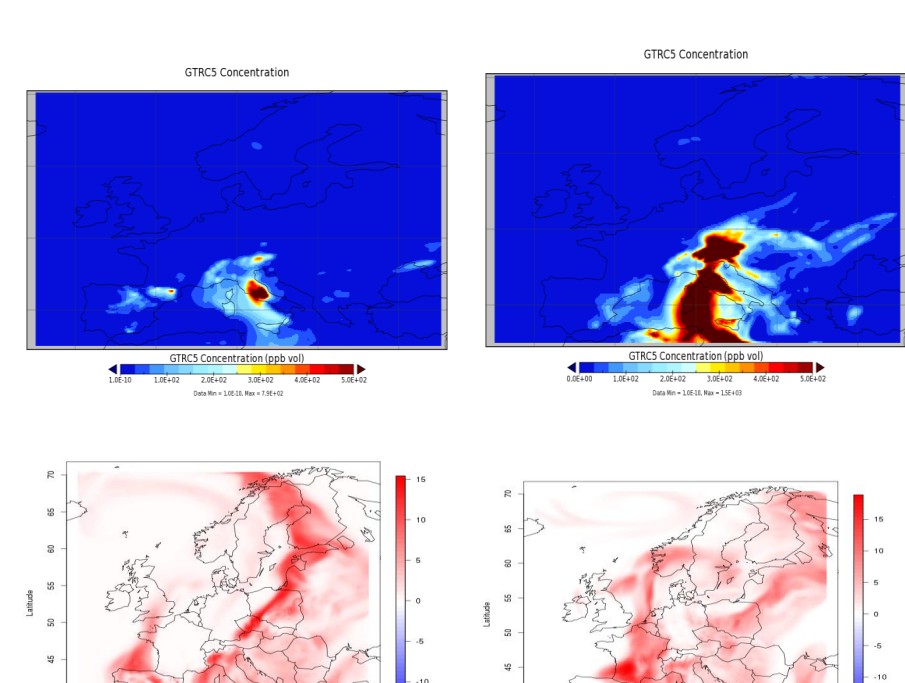

**Figure 8 (Upper panel): CHIMERE simulations of upper tropospheric tracer concentrations at 3 km for**
**April 26, 2008 (left column) and for April 27, 2008 (right column).**

**(Lower panel): CHIMERE simulations of photochemical production (difference between the reference**
**simulation and a simulation without emissions) at 3km (in ppb) for April 26, 2008 (left column) and for**
**April 27, 2008 (Right column).**





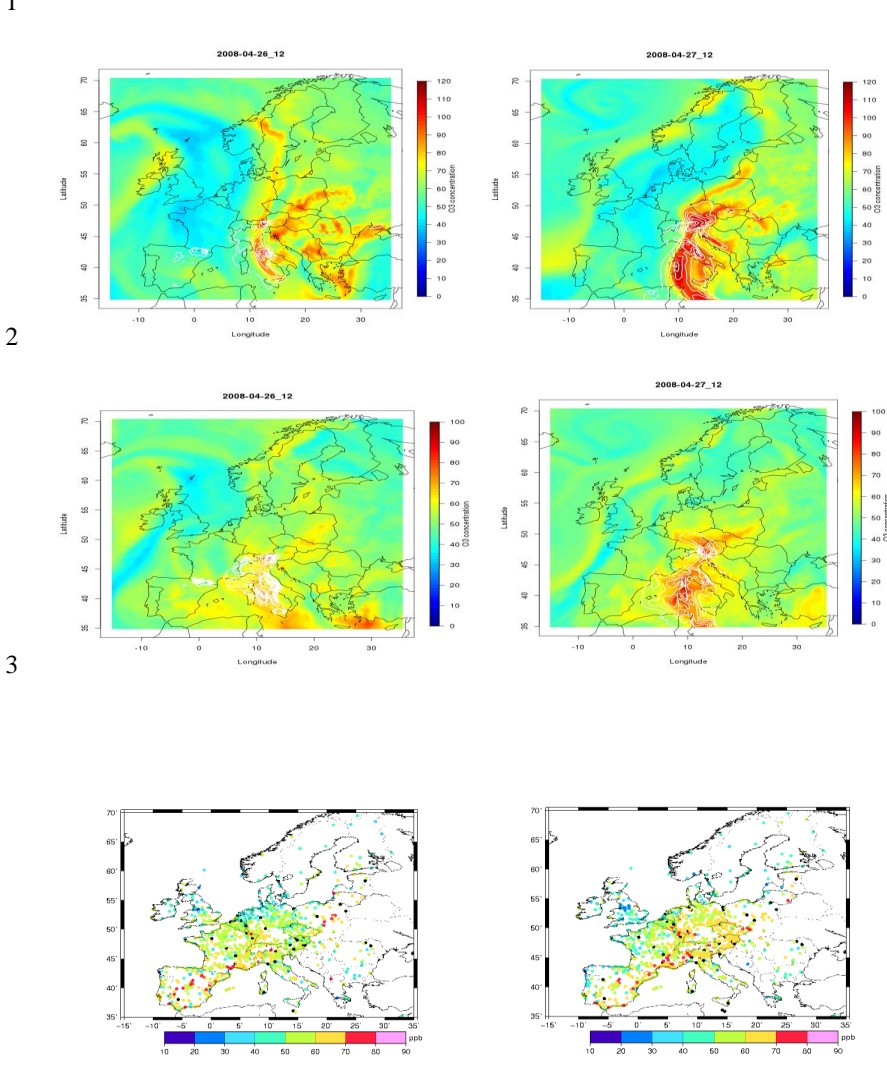

**Figure 9 (Upper panel):  CHIMERE simulations of the ozone field at 3 km altitude (in ppb) with the iso-**
**contours of the high tropospheric tracer (in white, arbitrary units) for April 26, 2008 (left column) and for**
**April 27, 2008 (right column).  (Middle panel): Same as in upper panel but for 1.5 km altitude. (Lower**
**panel): Hourly average surface ozone (EEA-AirBase) mixing ratios (ppb) at 15:00 h for April 26, 2008**
**(left column) and for April 27, 2008 (right column). Ozone data are from the EEA-AirBase database.**



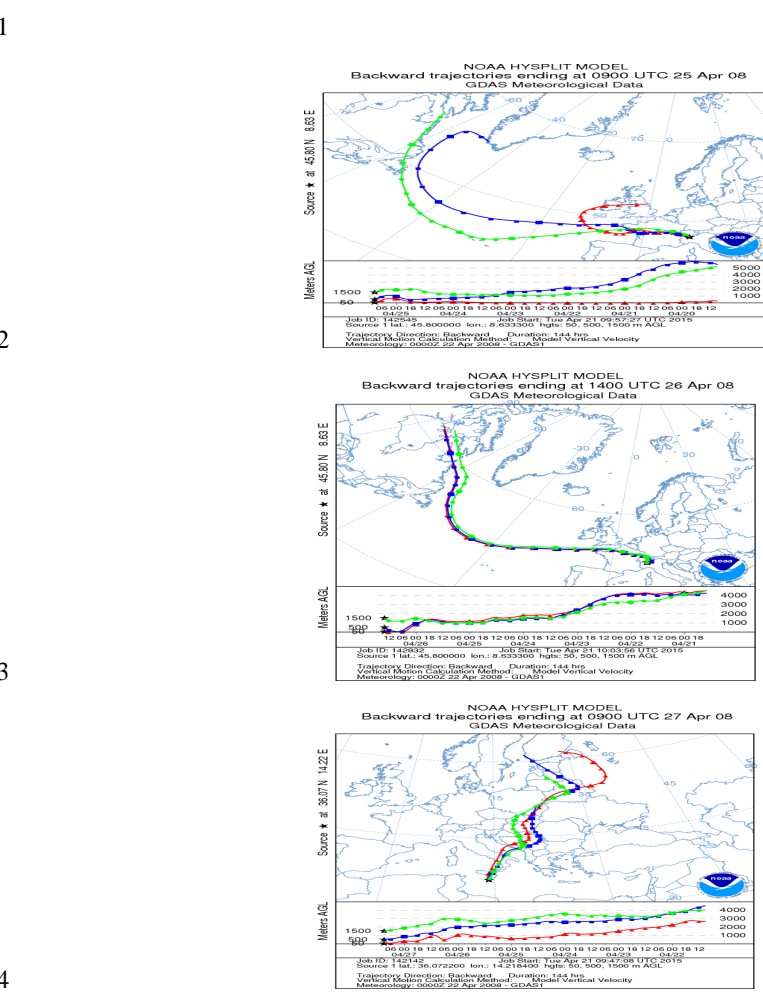

6 **Figure 10: Backward trajectories during the April 25-27 episode ending at EMEP rural ozone stations in**

7 **Italy (IT04, upper and middle panels) and Malta (MT01, lower panel).**



7    **Figure 11: Composite weather maps of geopotential height (left column) and specific humidity (right**

8    **column), for the high ozone episode of 7-9 May 2008 (lowest panels) as well as for two, three and five days**

9    **before.**



6      **Figure 12: Same as Fig. 8 but for omega vertical velocity (left column) and omega vertical velocity**

7      **anomaly (right column).**







6    **Figure 13: Same as Fig. 8 but for vector wind (left column) and air temperature anomaly (right column).**



5 **Figure 14: IASI satellite ozone measurements at 3km level (left column) and 10 km level (right column)**

6 **during the high ozone episode of 7-9 May 2008 (lowest panels) as well as for two, three and five days**

7 **before. Values outside the scale range are set up to the upper and lower color code respectively.**





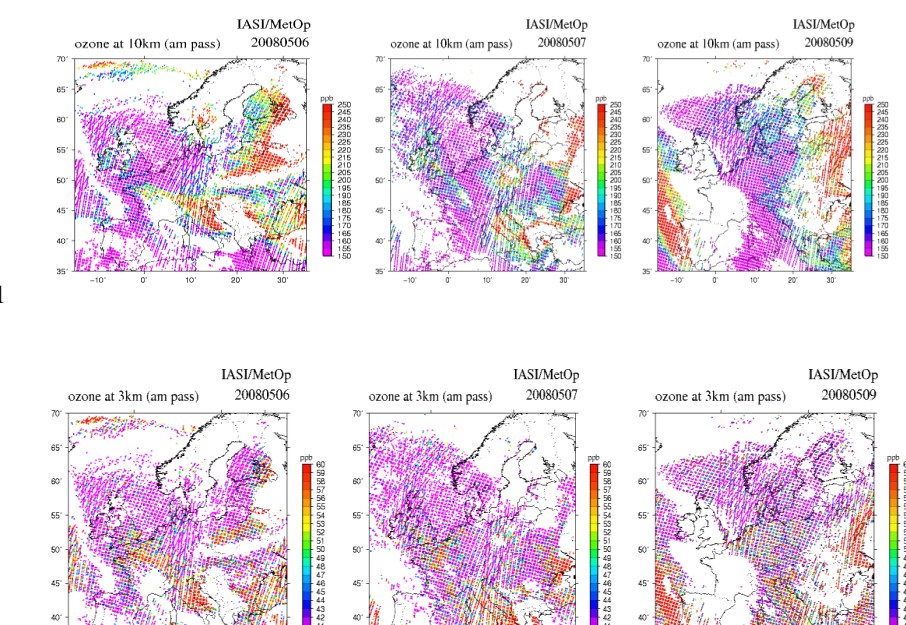

**Figure 15: Daily IASI satellite ozone measurements at 10 km level (upper panel) and at 3km level (lower**
**panel) for May 6, 2008 (left column), May 7, 2008 (middle column) and May 9, 2008 (right column).**
**Values outside the scale range are set up to the upper and lower color code respectively.**





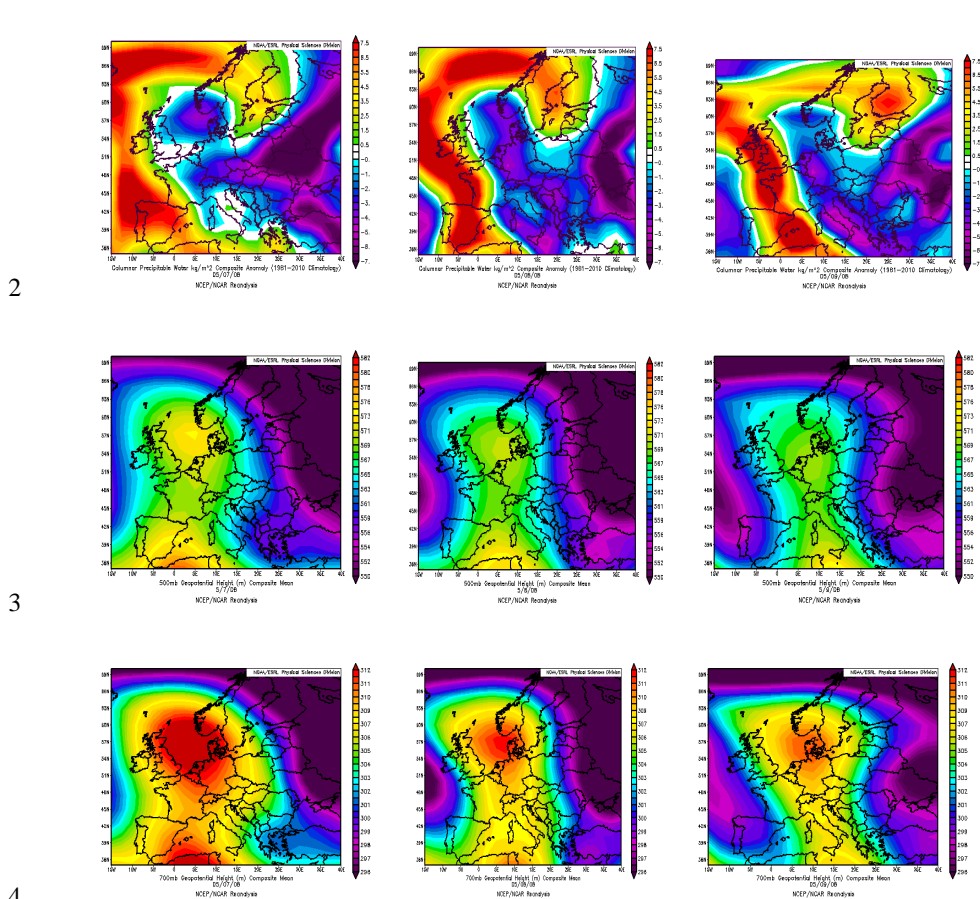

**Figure 16: Precipitable water anomaly (upper panel), geopotential height at 500hPa (middle panel) and**
**geopotential height at 700hPa (lower panel) for May 7, 2008 (left column), May 8, 2008 (middle column)**
**and May 9, 2008 (right column).**



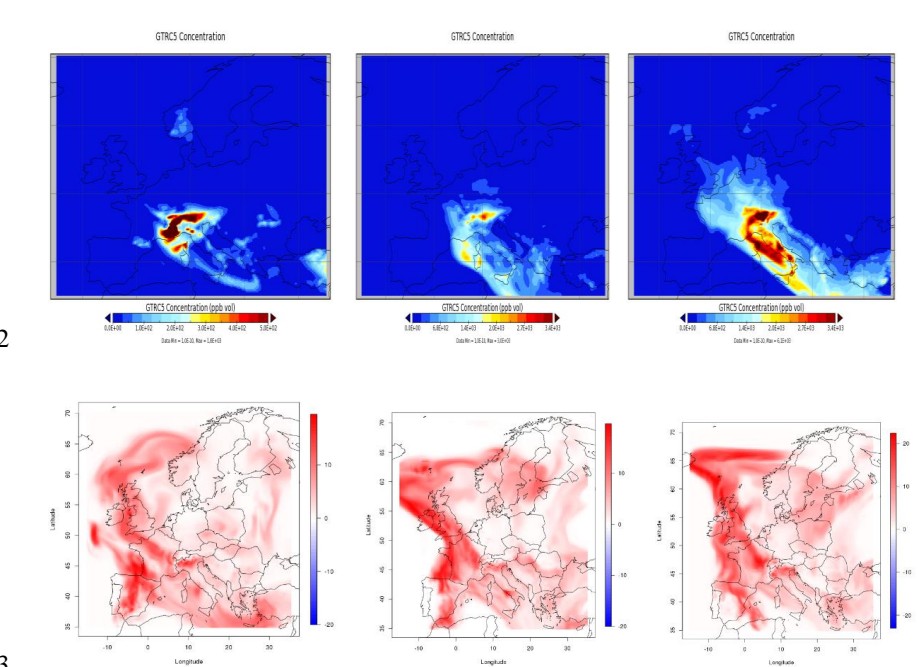

**Figure 17     (Upper panel): CHIMERE simulations of upper tropospheric tracer concentrations at 3 km**
**for May 7, 2008 (left), May 8, 2008 (middle) and for May 9, 2008 (right ).**
**(Lower panel): CHIMERE simulations of photochemical production (difference between the reference**
**simulation and a simulation without emissions) at 3km (in ppb) for May 7, 2008 (left), May 8, 2008**
**(middle) and for May 9, 2008 (right).**



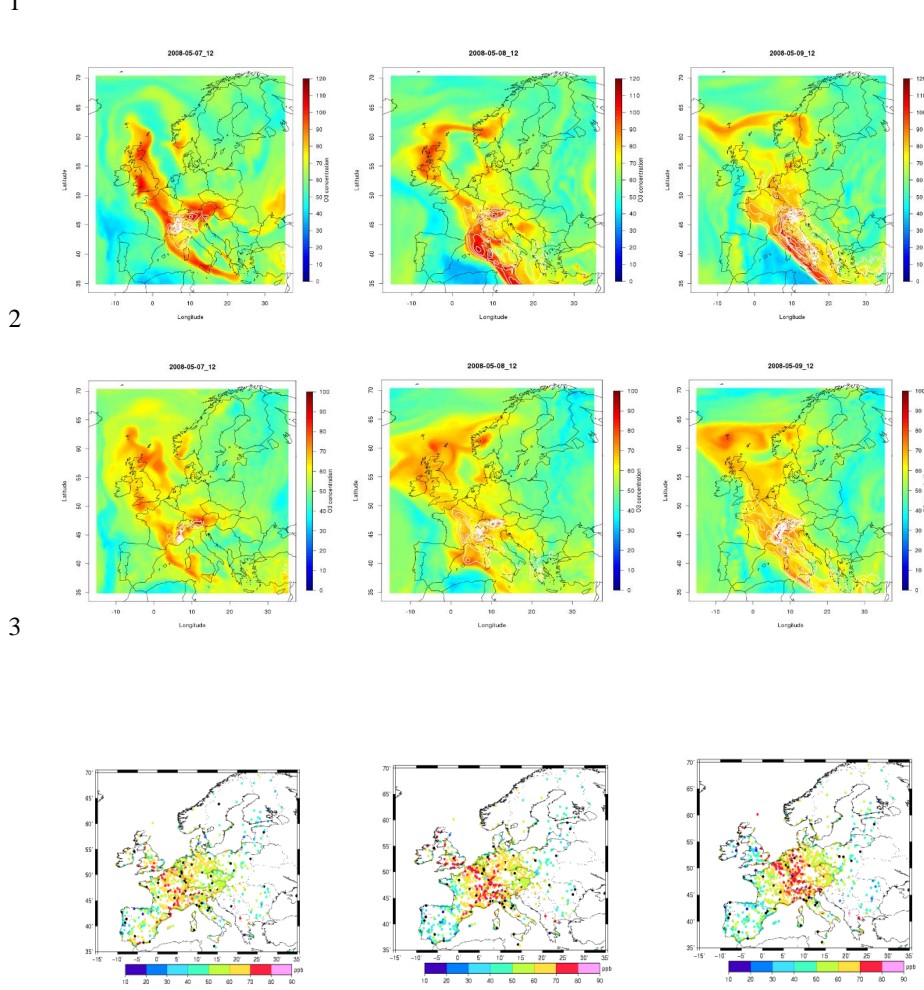

**Figure 18 (Upper panel): CHIMERE simulations of the ozone field at 3 km altitude (in ppb) with the iso-**
**contours of the high tropospheric tracer (in white, arbitrary units) for May 7, 2008 (left column), May 8,**
**2008 (middle column) and May 9, 2008 (right column). (Middle panel): Same as in upper panel but for 1.5**
**km altitude. (Lower panel): Hourly average surface ozone (EEA-AirBase) mixing ratios (ppb) at 15:00 h**
**for May 7, 2008 (left column), May 8, 2008 (middle column) and May 9, 2008 (right column). Ozone data**
**are from the EEA-AirBase database.**



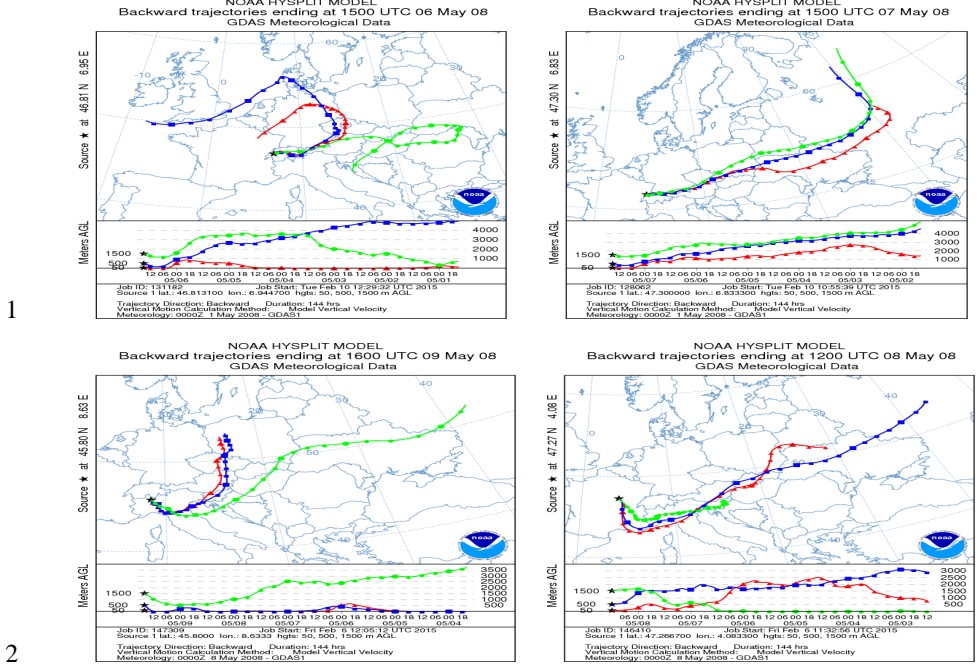

4   **Figure 19: Backward trajectories during the 6-9 May 2008 ozone episode ending at EMEP rural ozone**

5   **stations in Switzerland and Italy (left column) and France (right column).**





5 **Figure 20: Vertical profiles of ozone (red) and relative humidity (blue) over Payerne, Switzerland (left**

6 **column), Hohenpeissenberg, Germany (middle column) and Uccle, Belgium (right column) on May 7,**

7 **2008 (upper panel), May 8, 2008 (middle panel) and May 9, 2008 (lower panel).**





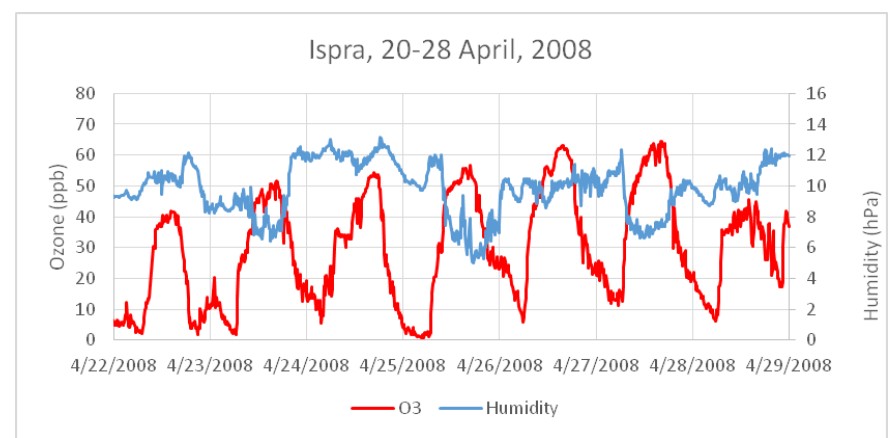

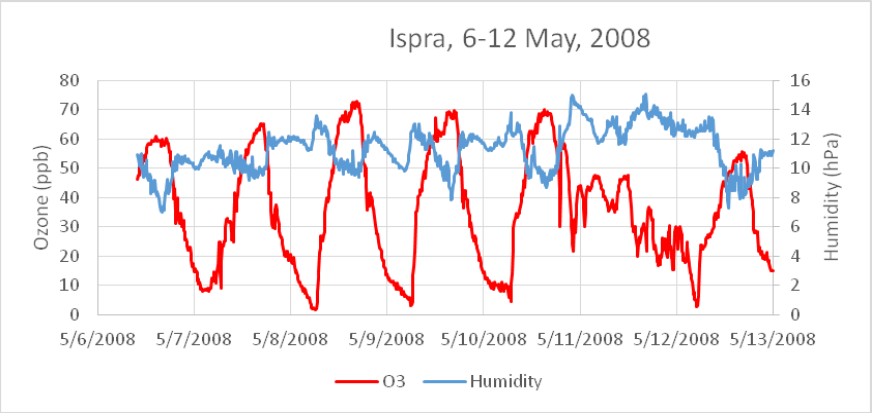

**Figure 21: (Upper panel): Ozone concentration (red) and absolute humidity (blue) measurements at the**
**JRC-Ispra station during the 26-27 April 2008 ozone episode.**
**(Lower panel): Ozone concentration (red) and absolute humidity (blue) measurements at the JRC-Ispra**
**station during the May 7-9, 2008 ozone episode.**