# Peer review of "An investigation on the origin of regional spring time ozone"

_Atmospheric Chemistry and Physics, 2016_

## Referee Comment (RC1) · Anonymous Referee #1 · 26 Sep 2016

From my initial read if the text, this manuscript has the potential to be an interesting paper in ACP, but unfortunately the presentation of the figures is so poor that I am unable to evaluate the analysis. The authors will have to completely revise and condense their figures before I can make a recommendation to the editor.

This paper has 21 figures with a total of 133 panels, 90% of which are too small to be read. This a result of the authors' poor formatting and visual display, coupled with ACPD's inadequate page layout which routinely makes figures far too small (I complain about this often). Any ACPD paper should be legible when printed onto paper, but for this paper, I can only make out the text in the figures by literally blowing up each panel on my laptop to 300% and then peering very hard to make sure I understand the highly

pixelated text. I am not prepared to review a paper such as this when there are over 100 tiny panels.

I also find the figure production to be below the standards of the journal. The authors did not produce the figures of the meteorological fields. All they did was go to the website of the NOAA/ESRL Physical Sciences Division and use the web-based plotting tool to make figures using coarse resolution NCEP reanalyses. While the figures from this website are fine for discussion of a paper when it is in draft form, they are not adequate for ACPD. The text is far too small and the color schemes are terrible and very difficult to interpret. This website also produces the data for each plot in a simple text file that can be used by the authors to plot the figures in a more visually appealing and understandable format, but they chose not to do so. Why did the authors choose to use the NCEP 2.5 degree data to investigate regional scale transport patters over the Mediterranean? As EU scientists these authors have free access to the superior ECMWF data, which are available at much higher resolution, far more appropriate to this regional scale study. The authors need to switch to ECMWF data and plot it with a resolution of at least 1 degree, preferably with a resolution of 0.5 degrees, if they are to properly understand the regional scale transport patterns. This is especially try for vertical velocity which is essentially useless at 2.5 degree resolution.

I have the same criticisms of the HYSPIT trajectories. The figures were not produced by the authors but simply taken from the HYSPLIT interactive webpage. When they were pasted into the manuscript they were shrunk vertically so that the aspect ratio is all wrong. The authors need to get the trajectory data from the website and plot the trajectories in a legible format.

I have similar complaints about the rest of the figures, except for figures 1, 20 and 21.

When the authors revise their figures they also need to greatly reduce the number of panels. No one wants to sit down with a paper and read through over 130 panels.

---

## Referee Comment (RC2) · Anonymous Referee #2 · 30 Sep 2016

The paper highlights one of the important factors, i.e. synoptic meteorological system,
controlling high ozone concentration episodes over Western Mediterranean (W-MED)
and Central Europe (C-EU), during spring. The text is generally well written and clear.
The time series plots of the observations clearly confirm there were two episodes in
which ozone builds up during late April and early May in 2008 over most of the Mediter-
ranean countries. However, when it comes to the proof of the argument, there are a
number of statements and images repeating the same messages. The analysis of var-
ious parameters that are generated from different types of observations and models
is a great idea, but unfortunately they are not always univocal. Therefore I think this
paper needs a major revision. Particularly, I would encourage the authors to avoid

misinterpreting the results of multiple sources.

The major concern I have with this paper, particularly in the result and discussion section, is that there is no clear focus on the two regions which are mentioned in the title (i.e. W-MED and C-EU). The NCEP/NCAR reanalysis maps (geopotential height, etc.) include enough evidences to confirm the existence of subsidence over those (two) regions during late April and early May, respectively. Also, there is a positive signal in all selected meteorological parameters on episode days; however the signal is not as strong as expected for some of the parameters. I cannot understand why the authors avoid focusing on them. Furthermore, the mechanisms leading to ozone enhancement as a consequence of high pressure systems should be explained in more detail. One mechanism could be the accumulation of surface ozone which is produced through chemical reactions due to the stagnant air flow. Another one could be linked to ozone flux from the upper troposphere to the surface, which the authors have already tried to prove but without sufficient arguments.

Detailed comments:

1) Abstract: "ozone measurement from countries surrounding Western Mediterranean..."_The promising title indicates that the study is focused on two different regions, W-MED and C-EU. Then in the abstract the focus changes to only one of them i.e. W-MED. Even the time series are plotted just for W-MED, why? According to the EEA-AirBase maps, Fig.18, the second episode is located over C-EU, isn't it? Wouldn't it be better to select a few stations from C-EU and plot their time series for them (as it has been done for W-MED)?

2) Abstract: "the results show that high ozone ..." _I think here you mean the results of the observation, don't you?

3) Abstract: "over these areas, strong..."_I think it is too much detail for the abstract section.

4) Introduction: in general the strongest focus of this part is over E-MED region during summer.

5) Introduction: "transport times are typically shorter..."_wouldn't it be better to rephrase this sentences to something like "they can be transported over longer distances than that in the boundary layer"?

6) Introduction: are there any references regarding the frequent existence of anticyclone condition over MED during spring?

7) Data and methodology: I think it would be nice to add the path (precise address) of the data center or website which the data are taken from.

8) Data and methodology: what is the horizontal and vertical resolution of the CHIMERE model?

9) Data and methodology: the description of methodology is insufficient.

10) Results and discussion, section 3.1: as it has been mentioned in this part, Fig. 1 (lower panel) shows a very good agreement between the results of different stations. So I cannot see the reason of having two different time series plots in Fig. 1?

11) Results and discussion, section 3.1: the title of this section is "... over Western Mediterranean", but the description mixes both episodes together. I would strongly recommend the authors to create a separate time series for each episode in each region as it has been done by airbase maps (lowest panel of Fig. 9 and Fig. 18). These maps and time series are enough proof for the confirmation of the existence of two episodes over these regions.

12) Results and discussion, section 3.2: since there are too many plots, I would suggest the authors to keep the plots related to the episode days in the main paper and move the others (i.e. the plots which they are related to the a few days before episodes) to the supplementary.

13) Results and discussion, section 3.2, a): is there any necessity to explain the low pressure system over another region i.e. E. Europe? How this system leads to high ozone concentration over W-MED?

14) Results and discussion, section 3.2, b): why do the authors describe negative specific humidity over Atlantic or etc.? The main focus of this part must be over W-MED and there is strong signal of negative specific humidity over this region (in the lowest panel of Fig. 2), why don't the authors concentrate on that?

15) Results and discussion, section 3.2, c): do we really need different maps of omega and its anomaly? In fact, both of them have the same messages.

16) Results and discussion, section 3.2, d): Yes, indeed there is a strong westerly wind toward W-MED a few days before the episode. {It may transfer ozone and its precursors from other places such as eastern US, etc. towards this region (via long-range transport), but there is not enough evidence for that through this maps.} However, on the 26th and 27th of April (in the lowest panel of the left column in Fig. 4), there is a weak wind flow over W-MED due to the existence of a high pressure system.

17) Results and discussion, section 3.2, e): in those episode days, there is a positive temperature anomaly over W-MED due to high pressure system. It leads to even more ozone production through photochemical reactions, doesn't it?

18) Results and discussion, section 3.2: "overall, over the same area of subsidence . . ."_how is it possible to have strong subsidence and strong wind together at the same time? Strong wind may even lead to a reduction of ozone by transferring them to the other regions.

19) Results and discussion, section 3.2: "In figure 5, the composite ozone IASI . . ."_ as the authors have already mentioned in this section, there is a signal of high ozone at free troposphere over C-MED and Atlantic. There is no explanation of how these are connected to high surface ozone over W-MED.

20) Results and discussion, section 3.2, page 9, first paragraph: I think, adding Fig. 7 is just overemphasizing the same messages which have been already explained in Fig. 1, 2, and 3.

21) Results and discussion, section 3.2, page 9, second paragraph: the CHIMERE simulation shows more or less the same results as IASI satellite data. There is high ozone in the free troposphere over C-MED. I cannot understand how this information is connected to high surface ozone over W-MED? I do not recommend the authors to apply a model simulation in this study without any evaluation of that.

22) More or less the same recommendations as above are valid for section 3.3.

23) Conclusions: "in this paper, the investigation of the regional . . ."_what does 'surrounding countries' mean? Does it mean C-EU?

24) Conclusions: paragraph 4: how do negative temperature anomaly and strong wind contribute to the high ozone level?

25) I would strongly recommend the authors to use a larger size for labels, title, etc. for all figures to make them readable.

Figure 1: The lower time series plot clearly shows the episodes. There is no need to keep both plots. Furthermore, the unit of ozone in the legend should be "ppb" instead of "ug/m3", shouldn't it?

Figure 2: The right column is specific humidity anomaly, but in the legend it is written specific humidity. It would be recommendable to add the unit of this parameter.

Figure 3: Both omega and omega anomaly have the same messages; I would recommend the authors to keep only one of them. Units are missing.

Figure 4: Again, adding units would be recommendable.

Figure 9: It is hard to see the white contour over CHIMERE maps. In the legend the color of contours is labeled black instead of white. I do not know what the aim is of

putting surface ozone (from EEA-AirBase) maps separately below CHIMERE simulation maps.

[Figure]

---

## Short Comment (SC1) · 11 Oct 2016

The paper by Kalabokas et al. strongly relies on ozone observations from various measurement networks and platforms but none of the data providers is properly accredited in the acknowledgements. I am part of the operators team of the Swiss National Air Pollution Monitoring Network – which is one of the providers of the EMEP data used here, namely Payerne (CH02) and Chaumont (CH04), even though I am not directly involved in any of the used observations. In other words, I am not writing to claim any personal credits. However, our network's funding partly relies on a sound annual reporting proofing the interest in our observations by the (scientific) community. Therefore, a proper acknowledgement in publications using our data is vital for the long-term operation of

the measurements. This is also true for many other monitoring networks. As host of the WMO/GAW Quality Assurance / Science Activity Centre Switzerland (QA/SAC-CH), we also continuously promote the timely submission of data to the international data repositories. Missing credit for data made publicly available is one of the main arguments against a (fast) provision of the time series. Thus, the missing declaration of the data source will also jeopardize similar analysis in the long run. Moreover, the EMEP/EBAS data policy accessibly on the EBAS webpage clearly reads: "For scientific purposes, access to these data is unlimited and provided without charge. By their use you accept that an offer of co-authorship will be made through personal contact with the data providers or owners whenever substantial use is made of their data. In all cases, an acknowledgement must be made to the data providers or owners and to the project name when these data are used within a publication."

---

## Author Comment (AC1) · 2 Dec 2016

**Reviewer 1 Comment**

From my initial read if the text, this manuscript has the potential to be an interesting paper in ACP, but unfortunately the presentation of the figures is so poor that I am unable to evaluate the analysis. The authors will have to completely revise and condense their figures before I can make a recommendation to the editor.

This paper has 21 figures with a total of 133 panels, 90% of which are too small to be read. This a result of the authors' poor formatting and visual display, coupled with ACPD's inadequate page layout which routinely makes figures far too small (I complain about this often). Any ACPD paper should be legible when printed onto paper, but for this paper, I can only make out the text in the figures by literally blowing up each panel on my laptop to 300% and then peering very hard to make sure I understand the highly pixelated text. I am not prepared to review a paper such as this when there are over 100 tiny panels.

I also find the figure production to be below the standards of the journal. The authors did not produce the figures of the meteorological fields. All they did was go to the website of the NOAA/ESRL Physical Sciences Division and use the web-based plotting tool to make figures using coarse resolution NCEP reanalyses. While the figures from this website are fine for discussion of a paper when it is in draft form, they are not adequate for ACPD. The text is far too small and the color schemes are terrible and very difficult to interpret. This website also produces the data for each plot in a simple text file that can be used by the authors to plot the figures in a more visually appealing and understandable format, but they chose not to do so. Why did the authors choose to use the NCEP 2.5 degree data to investigate regional scale transport patters over the Mediterranean? As EU scientists these authors have free access to the superior ECMWF data, which are available at much higher resolution, far more appropriate to this regional scale study. The authors need to switch to ECMWF data and plot it with a resolution of at least 1 degree, preferably with a resolution of 0.5 degrees, if they are to properly understand the regional scale transport patterns. This is especially try for vertical velocity which is essentially useless at 2.5 degree resolution.

I have the same criticisms of the HYSPIT trajectories. The figures were not produced by the authors but simply taken from the HYSPLIT interactive webpage. When they were pasted into the manuscript they were shrunk vertically so that the aspect ratio is all wrong. The authors need to get the trajectory data from the website and plot the trajectories in a legible format.

I have similar complaints about the rest of the figures, except for figures 1, 20 and 21. When the authors revise their figures they also need to greatly reduce the number of panels. No one wants to sit down with a paper and read through over 130 panels.

**Authors Response (in Italics)**

*We would like to thank the reviewer for his comments, which help improving the paper.*

*At first, we would like to express our surprise for the strong criticism of the reviewer towards NOAA/ESRL charts as we had already used them in the same form previously, without any objection, in at least 4 papers, which are included in the list of references: 2 in Atmospheric Chemistry and Physics (Kalabokas et al., 2007; 2013), 1 in Atmospheric Environment (Kalabokas et al., 2008) and 1 in Tellus B (Kalabokas et al., 2015).*

*Following the reviewer's suggestion, we tried to replace the NOAA/ESRL maps to the corresponding ECMWF maps. It appears though that the ECMWF web-page does not offer corresponding possibilities for producing composite charts. So, we relied to software developed at the LISA-Paris laboratory for the production of Figures based on ECMWF data.*

*It turned out also that we could not produce temperature anomalies and specific humidity anomalies based on ECMWF data. It was possible to produce composite charts of geopotential height, wind speed and omega at the 900 and 800 hPa pressure levels, shown in new Figs 3 and 8 for the April and May episodes respectively. In the new Figs 2 and 7 the composite charts of all examined meteorological parameters are presented at the 850hPa pressure level, based on NOAA/ESRL data, but they were plotted by using the same plotting tool as for the ECMWF maps, which was the initial suggestion of the reviewer. As observed, both chart types (ECMWF and NOAA/ESRL) show more or less the same patterns, which suggests that for the examination of high ozone episodes the choice of meteorological map type does not influence the used argumentation.*
*In any case, the original NOAA/ESRL plotting format does not appear in the main Figure list, according to the reviewer's suggestion..*

*We also improved the layout of the HYSPLIT back-trajectories by using the maximum available plot resolution from the web-page, since we do not dispose another alternative. We also corrected the aspect ratio of these charts following the reviewer's suggestion.*

*We also proceeded to a substantial reduction of Figs, as it was suggested by the reviewer. With these arrangements the number of Figs in the Figure list is reduced to 11 and the number of panels is reduced to 52 (more than 50 % reduction). We also decreased the number of panels per page for easier reading. On the other hand, we just created an Annex (as it was also suggested by reviewer 2) for shifting there some of the Figs, which might be interesting for some readers.*

*Following the above arrangements the paper has been restructured, as seen in the attached revised version, and we believe that the reviewer's requirements are now fully fulfilled.*
*The corresponding changes in the revised manuscript are highlighted in yellow color.*

---

## Author Comment (AC2) · 2 Dec 2016

**Reviewer 2 Comment**

The paper highlights one of the important factors, i.e. synoptic meteorological system, controlling high ozone concentration episodes over Western Mediterranean (W-MED) and Central Europe (C-EU), during spring. The text is generally well written and clear. The time series plots of the observations clearly confirm there were two episodes in which ozone builds up during late April and early May in 2008 over most of the Mediterranean countries. However, when it comes to the proof of the argument, there are a number of statements and images repeating the same messages. The analysis of various parameters that are generated from different types of observations and models is a great idea, but unfortunately they are not always univocal. Therefore I think this paper needs a major revision. Particularly, I would encourage the authors to avoid misinterpreting the results of multiple sources.

The major concern I have with this paper, particularly in the result and discussion section, is that there is no clear focus on the two regions which are mentioned in the title (i.e. W-MED and C-EU). The NCEP/NCAR reanalysis maps (geopotential height, etc.) include enough evidences to confirm the existence of subsidence over those (two) regions during late April and early May, respectively. Also, there is a positive signal in all selected meteorological parameters on episode days; however the signal is not as strong as expected for some of the parameters. I cannot understand why the authors avoid focusing on them. Furthermore, the mechanisms leading to ozone enhancement as a consequence of high pressure systems should be explained in more detail. One mechanism could be the accumulation of surface ozone which is produced through chemical reactions due to the stagnant air flow. Another one could be linked to ozone flux from the upper troposphere to the surface, which the authors have already tried to prove but without sufficient arguments.

**Authors Response (in Italics)**

*We would like to thank the reviewer for his comments, which help improving the paper. We think that it would be more practical to structure our response, presented below, according to the very detailed comments of the reviewer.*
*The corresponding changes in the revised manuscript are highlighted in green color.*

Detailed comments:

1) Abstract: "ozone measurement from countries surrounding Western Mediterranean. . ."_The promising title indicates that the study is focused on two different regions, W-MED and C-EU. Then in the abstract the focus changes to only one of them i.e. W-MED. Even the time series are plotted just for W-MED, why? According to the EEA-AirBase maps, Fig.18, the second episode is located over C-EU, isn't it? Wouldn't it be better to select a few stations from C-EU and plot their time series for them (as it has been done for W-MED)?

*In fact the region of study is the W-MED and the corresponding EMEP stations in that region have been selected for both episodes with highest ozone values over the region. At a later stage of the analysis it came out that especially for the May episode high ozone levels have been also recorded at Central Europe at the same time and for that reason this region was added also to the title. We could add another Figure presenting measurements from some selected Central European stations but this would be in conflict with the remarks of Reviewer 1 who suggests to reduce substantially the number of Fig and which will be reduced to the half of their original number. In addition, the presented EMEP stations in Switzerland and France cover also some parts of the central European domain. For this purpose, also, we will add in the text the stations names as well as geographical coordinates. As a final precaution and for avoiding confusion, we will remove form the title "Central Europe" as the paper is essentially concentrating on the W-MED.*

2) Abstract: "the results show that high ozone . . ." _I think here you mean the results of the observation, don't you?

*Yes, indeed. The phrase will be modified accordingly ("results" will be replaced by "observations").*

3) Abstract: "over these areas, strong..."_I think it is too much detail for the abstract section.

*This part of the abstract will be reduced according to the reviewer's suggestion by removing 3 lines from the abstract text ("Over these areas……………500hPa pressure levels").*

4) Introduction: in general the strongest focus of this part is over E-MED region during summer.

*Yes, indeed. As mentioned, our initial focus was the Mediterranean basin where the highest ozone observations are observed in its Eastern part during summer and, in fact, a major finding of this paper is that comparable synoptic meteorological conditions exist between EMED in summer and WMED in spring during ozone episodes. According to the reviewer's suggestion, some more elements on previous studies on the WMED will be added.*

5) Introduction: "transport times are typically shorter. . ."_wouldn't it be better to rephrase this sentences to something like "they can be transported over longer distances than that in the boundary layer"?

*This phrase will be modified according to the reviewer's suggestion.*

6) Introduction: are there any references regarding the frequent existence of anticyclone condition over MED during spring?

*When examining the average seasonal climatological charts of geopotential heights, it is clear that the N. African anticyclone is progressively extending and moving towards the Central Mediterranean when passing from winter to spring and summer months (http://www.esrl.noaa.gov/psd/cgi-bin/data/composites/printpage.pl).*

 7) Data and methodology: I think it would be nice to add the path (precise address) of the data center or website which the data are taken from.

*The precise address of the data center or website from which the data are taken will be added in the text, according to the reviewer's suggestion (http://www.nilu.no/projects/ccc/onlinedata/ozone/).*

8) Data and methodology: what is the horizontal and vertical resolution of the CHIMERE model?

*The current configuration of the CHIMERE model uses a horizontal resolution of 0.25° x 0.25° and 30 hybrid (σ,p) vertical levels are used to describe the whole troposphere (i.e from the ground to 200hPa).*

9) Data and methodology: the description of methodology is insufficient.

*An effort will be made to improve the methodology section and for that purpose a sentence will be added at the end of this section ("The satellite.....boundary layer").*

10) Results and discussion, section 3.1: as it has been mentioned in this part, Fig. 1 (lower panel) shows a very good agreement between the results of different stations. So I cannot see the reason of having two different time series plots in Fig. 1?

*Just to show that the regional ozone episodes affect more WMED countries (Spain +Malta), covering most of the basin.  It would be difficult to show that without Fig.1a. Especially for the April episode, Malta shows the highest ozone concentrations over several days, attributed mainly to tropospheric transport.*

11) Results and discussion, section 3.1: the title of this section is ". . . over Western Mediterranean", but the description mixes both episodes together. I would strongly recommend the authors to create a separate time series for each episode in each region as it has been done by airbase maps (lowest panel of Fig. 9 and Fig. 18).

These maps and time series are enough proof for the confirmation of the existence of two episodes over these regions.

*At first, as mentioned also above, "Central Europe" will be removed from the title as the paper is mainly focused on the Western Mediterranean ozone episodes although Central Europe is in fact also influenced at the same time, especially during the May episode. The selected April and May episodes seem to be the most characteristic of the season but we think that it is worth it to show also the other high (or low) ozone episodes occurring during spring 2008 and showing that mid-day ozone concentrations might have the same variation pattern even at large distances between rural ozone stations, which indicates regional episodes. In addition, due to strong objections from reviewer 1 we need to reduce substantially the number of Figs, which have been reduced by about the half of their initial number.*

12) Results and discussion, section 3.2: since there are too many plots, I would suggest the authors to keep the plots related to the episode days in the main paper and move the others (i.e. the plots which they are related to the a few days before episodes) to the supplementary.

*It will be done so, following also corresponding remarks of reviewer 1, as about half of the plots will be moved to the Annex.*

13) Results and discussion, section 3.2, a): is there any necessity to explain the low pressure system over another region i.e. E. Europe? How this system leads to high ozone concentration over W-MED?

*In fact one of the main points of the paper is that the interaction of high and low pressure systems create conditions of subsidence, especially at the interface of both meteorological systems. So, the extent and the intensity of high and low pressure systems are very important for a better understanding of this phenomenon. A key point is that the WMED area is influenced by this process as the air masses arriving there originate from the subsidence area located in E. Europe.*

14) Results and discussion, section 3.2, b): why do the authors describe negative specific humidity over Atlantic or etc.? The main focus of this part must be over WMED and there is strong signal of negative specific humidity over this region (in the lowest panel of Fig. 2), why don't the authors concentrate on that?

*Essentially for the same reasons as in the previous remark. The extended subsidence over the Atlantic affects WMED through transport as back-trajectories and the meteorological charts show.*

15) Results and discussion, section 3.2, c): do we really need different maps of omega and its anomaly? In fact, both of them have the same messages.

*We wanted to put more emphasis that during the ozone episodes, unusual vertical exchange conditions occur (positive omega, indicating subsidence). The omega anomalies could be removed from the main Figure list, also in the spirit of the remarks of reviewer 1 for reducing the total number of Figs.*

16) Results and discussion, section 3.2, d): Yes, indeed there is a strong westerly wind toward W-MED a few days before the episode. {It may transfer ozone and its precursors from other places such as eastern US, etc. towards this region (via long-range transport), but there is not enough evidence for that through this maps.} However, on the 26th and 27th of April (in the lowest panel of the left column in Fig. 4), there is a weak wind flow over W-MED due to the existence of a high pressure system.

*The back-trajectories in combination with the IASI satellite measurements show that the flow over W-MED originates from the high tropospheric ozone area over N. Atlantic. Of course, photochemical ozone production inside the anticyclone might also occur, which will be more emphasized in the paper.*

17) Results and discussion, section 3.2, e): in those episode days, there is a positive temperature anomaly over W-MED due to high pressure system. It leads to even more ozone production through photochemical reactions, doesn't it?

*Yes, of course. But the simultaneous appearance of positive and negative temperature anomalies is also a sign of tropospheric processes leading to subsidence occurring at the interface of both areas but shifted somewhat towards the negative temperature anomalies area, which is associated with colder and richer in ozone tropospheric air masses.*

18) Results and discussion, section 3.2: "overall, over the same area of subsidence . . ."_how is it possible to have strong subsidence and strong wind together at the same time? Strong wind may even lead to a reduction of ozone by transferring them to the other regions.

*This is actually a very good point, which might help clarifying the discussion. As mentioned also above, during high ozone episodes, we might have strong downward transport (or subsidence) and strong winds at the same time, which usually originate from high tropospheric ozone reservoirs (a frequently occurring situation observed at the eastern Mediterranean during summertime). This is, in fact one of the main points of this paper: during high springtime ozone episodes in the WMED comparable synoptic conditions and atmospheric processes occur as during high ozone in the EMED in summertime. On the contrary, lower ozone levels occur in the troposphere during autumn and winter seasons under similar conditions, as the tropospheric ozone levels are significantly lower. In addition, in EMED during summer strong westerly winds, transporting boundary layer air from the Atlantic, are usually associated with low ozone but the corresponding synoptic conditions are quite different if compared with the high ozone episodes associated with strong northerly descending winds over the Aegean Sea and the EMED (Kalabokas et al., 2013; Kalabokas et al., 2015).*

19) Results and discussion, section 3.2: "In figure 5, the composite ozone IASI . . ."_ as the authors have already mentioned in this section, there is a signal of high ozone at free troposphere over C-MED and Atlantic. There is no explanation of how these are connected to high surface ozone over W-MED.

*As also mentioned previously and according to observations, the connection might occur through advection, which follows the subsidence observed over the Atlantic. In fact, the CMED high ozone maximum observed by IASI is associated with processes occurring within the low-pressure system, leading to the enhancement of ozone levels. We agree that according to the meteorological analysis no influence is observed from the CMED tropospheric ozone maximum to the surface ozone observations in WMED.*

20) Results and discussion, section 3.2, page 9, first paragraph: I think, adding Fig. 7 is just overemphasizing the same messages which have been already explained in Fig. 1, 2, and 3.

*The figure will be removed following the reviewer's suggestion and will be put in the Annex. The idea is just to show that similar synoptic patterns and associated processes occur also at 700hPa and 500 hPa levels, which indicate deep subsidence throughout the troposphere.*

21) Results and discussion, section 3.2, page 9, second paragraph: the CHIMERE simulation shows more or less the same results as IASI satellite data. There is high ozone in the free troposphere over C-MED. I cannot understand how this information is connected to high surface ozone over W-MED? I do not recommend the authors to apply a model simulation in this study without any evaluation of that.

*As mentioned previously, the analysis shows that surface ozone in W-MED is influenced from the high ozone reservoir over the Atlantic region (in addition to the photochemical production during the last days of the episode). On the contrary, we agree that the high tropospheric ozone over C-MED (related to the low pressure system to the east) does not influence the surface ozone concentrations in W-MED, as also mentioned above.*

22) More or less the same recommendations as above are valid for section 3.3.

*The May episode described in section 3.3 is quite different than the April episode regarding synoptic meteorological conditions, as the main anticyclone associated with the ozone episode is located in central and northern Europe and the corresponding discussion concerning the synoptic influence on ozone levels has been adopted accordingly.*

23) Conclusions: "in this paper, the investigation of the regional . . ."_what does 'surrounding countries' mean? Does it mean C-EU?

*The phrase means 'surrounding countries' of the northern and eastern part of the western Mediterranean basin, from which the results of the EMEP stations are presented in Fig. 1 (Spain, France, Switzerland, Italy, Malta).*

24) Conclusions: paragraph 4: how do negative temperature anomaly and strong wind contribute to the high ozone level?

*As mentioned also previously, strong winds might be associated with high ozone if they originate from a high ozone reservoir located in the upper tropospheric layers (as IASI measurements indicate). This rapid downward transport, transporting colder and richer in ozone air from upper layers located to the north, is associated in fact with a negative temperature anomaly, which usually appears next to a positive temperature anomaly area, located inside the anticyclone during subsidence conditions. These observations are in agreement and support the corresponding observations from the analysis of other meteorological parameters, indicating strong vertical tropospheric transport originating from northern directions.*

25) I would strongly recommend the authors to use a larger size for labels, title, etc. for all figures to make them readable.

*The new ECMWF Figs will take into account the above remarks.*

Figure 1: The lower time series plot clearly shows the episodes. There is no need to keep both plots. Furthermore, the unit of ozone in the legend should be "ppb" instead of "ug/m3", shouldn't it?

*The figure just shows that the high ozone concentrations occur also at Spain and Malta, especially for the April episode, which is important to show its extend over the western Mediterranean basin.*
*Yes, indeed, "ppb" should be used instead of "ug/m3", we apologize for this mistake.*

Figure 2: The right column is specific humidity anomaly, but in the legend it is written specific humidity. It would be recommendable to add the unit of this parameter.

*In fact, "anomaly" should be added to "specific humidity". Also, the unit of this parameter will be added in the legend.*

Figure 3: Both omega and omega anomaly have the same messages; I would recommend the authors to keep only one of them. Units are missing.

*As mentioned also above, only omega will be retained in the main Figure list.*

Figure 4: Again, adding units would be recommendable.

*Units will be added in the legend, according to reviewer's suggestion.*

Figure 9: It is hard to see the white contour over CHIMERE maps. In the legend the color of contours is labeled black instead of white. I do not know what the aim is of putting surface ozone (from EEA-AirBase) maps separately below CHIMERE simulation maps.

*The white tracer contours of the CHIMERE maps will be replaced by black ones.  Also the EEA-Airbase observations will be put together with the IASI measurements in the main Figure list (new Figs 3, 7).*

---

## Author Comment (AC3) · 2 Dec 2016

**Interactive comment, 3.**

The paper by Kalabokas et al. strongly relies on ozone observations from various measurement networks and platforms but none of the data providers is properly accredited in the acknowledgements. I am part of the operators team of the Swiss National Air Pollution Monitoring Network – which is one of the providers of the EMEP data used here, namely Payerne (CH02) and Chaumont (CH04), even though I am not directly involved in any of the used observations. In other words, I am not writing to claim any personal credits. However, our network's funding partly relies on a sound annual reporting proofing the interest in our observations by the (scientific) community. Therefore, a proper acknowledgement in publications using our data is vital for the long-term operation of the measurements. This is also true for many other monitoring networks. As host of the WMO/GAW Quality Assurance / Science Activity Centre Switzerland (QA/SACCH), we also continuously promote the timely submission of data to the international data repositories. Missing credit for data made publicly available is one of the main arguments against a (fast) provision of the time series. Thus, the missing declaration of the data source will also jeopardize similar analysis in the long run. Moreover, the EMEP/EBAS data policy accessibly on the EBAS webpage clearly reads: "For scientific purposes, access to these data is unlimited and provided without charge. By their use you accept that an offer of co-authorship will be made through personal contact with the data providers or owners whenever substantial use is made of their data. In all cases, an acknowledgement must be made to the data providers or owners and to the project name when these data are used within a publication."

**Authors Response (in Italics)**

*We really apologize for the lack of particular acknowledgement to the operators of the air quality stations, thinking that mentioning the EMEP programme and its data platform would be sufficient. We agree totally with the spirit of this comment. So, for that purpose, we will add at first a reference for the EMEP program (EMEP, Chemical Coordinated Center, www.emep.int) as well as the station names and their geographical coordinates in the manuscript text (Data and Methodology section).*
*Finally, we will add the following sentence at the end of acknowledgements. "Our particular thanks go to the operators of the following air quality stations: Cabo de Creus in Spain, Morvan and Montandon in France, Payerne and Chaumont in Switzerland, Montelibretti and Ispra in Italy and Giordan Lighthouse in Malta".*

*The corresponding changes in the revised manuscript are highlighted in blue color.*

---

## Author Response (AR2)

Co-Editor Decision: Reconsider after major revisions (02 Jan 2017) by Timothy Bertram
Comments to the Author:
Dear Pavlos-

Having received additional comments on your manuscript, it is my decision that major revisions are still required for publication in ACP. These revisions are specifically related to the figure quality. Please consult with the ACP production team if you have questions or concerns regarding figure quality, resolution, etc.

*Following the Co-editor's advice, we consulted the ACP production team so that the updated version of Figs complies fully with the ACP standards*

Non-public comments to the Author:
Dear Pavlos-

I have received comments on your revised paper. While I understand that the figure types/quality in this manuscript have been used for prior publications, it is the reviewers (and my opinion) that higher quality graphics would substantially aid in conveying the messages carried in this paper. I would be happy to review the paper again if the following changes to the figures are considered. I understand the time investment, but in the paper's current form, I think much of the discussion and many of the conclusions are not seen in the figures. I will review the scientific merits of the paper again once the following edits are made.

1) Figure resolution: Many of the map figures are too small to be seen without blowing them up and the figures are basically illegible when printed. Figures should be provided as crisp high-resolution tiff or postscript images. As a guide, when the manuscript is printed, all important elements of the figure should be legible to the reader.

2) The pages are full of white space with tiny panels. Place the colorbars underneath the panels and fill the page from the left margin to the right margin with the panels, with just a couple of millimeters between panels. Efficient use of the page will permit the reader to see the figures.

*As mentioned above, we consulted the ACP production team regarding the size of Figs, so that the updated version with magnified Figs complies fully with the ACP standards.*

3) The wind barbs are difficult to see because they are the same color as the map. White wind barbs are more obvious.

*Following the reviewer's suggestion we replaced the black arrows for wind velocity with white ones in Figs 3 and 8, although reviewer 2 had suggested the opposite (replacing white contours with black ones) in the first version of the manuscript submitted last July (Figs 9 and 18).*

4) The color scale of the IASI images is not very helpful. Having bright magenta (low ozone) very close to bright, bright red (high ozone) is not intuitive for grasping mixing ratio gradients.

*As mentioned above, we consulted the ACP production team regarding the color code of Figs so that the updated version of Figs complies fully with the ACP standards.*

5) The 10 km IASI figures are meant to tell me about ozone in the upper troposphere. But the images are provided with a stratospheric color scale with the lowest value being 150 ppb. For me to understand ozone in the UT and the transition to the stratosphere I need a color scale that covers the range 30-150 ppbv. If the coarse vertical resolution of IASI does not effectively detect ozone values at 10 km below 150 ppb, then this is the wrong product to use and another product should be used. OMI provides ozone in the UT in the 300 hPa range.

*Our opinion is that IASI measurements are an appropriate tool for the ozone detection in the upper troposphere as well as to the transition to the stratosphere and IASI is perfectly able to detect ozone values below 150 ppb. In fact, we have tried with various scales at both tropospheric levels (3 and 10km). In the following panels we show examples of some scales that we tested for 10 km for the April episode (daily maps as well as composite maps). As mentioned in the text the ozone distribution follows quite well the synoptic atmospheric patterns (high ozone over the low pressure systems, low ozone over the anticyclone). In fact, the scale presented in the manuscript (150-250 ppb) was selected because it presents better the ozone distribution and the concentration gradients observed in the upper troposphere (including possible stratospheric influence) during the examined episodes.*

[Figure]

*In addition, it should be also mentioned that IASI vertical profiles or distributions over the Mediterranean or over Asia are presented in the following papers (included in the reference list of the manuscript): (Dufour et al., 2012; Doche et al., 2014; Dufour et al., 2015).*

6) Figure 7 repeats the specific humidity anomaly panel and omits the vertical velocity.

*The specific humidity anomaly in Fig. 7 (lower panel) was replaced by the appropriate vertical velocity panel and we apologize for this mistake.*

7) Why use both NCEP and ECMWF? It is suggested that ECMWF be used as these products are of better quality. If both are used, a strong justification should be included.

*As we had already mentioned, we tried to do our best to satisfy the comments of reviewer 1 regarding the replacement of NOAA/ESRL maps by the ECMWF maps. So, we produced the corresponding ECMWF maps, by using our own plotting infrastructure (LISA-Paris) which had to be specifically adjusted, for the parameters that we could get from ECMWF (geopotential height, wind speed and vertical velocity) and for the available pressure levels (900 and 800 hPa levels). For the remaining parameters (temperature anomaly and specific humidity anomaly) we plotted new Figs by our own plotting means by using the numerical files provided by NOAA/ESRL at the available pressure level (850 hPa). We think that this combination has the advantage of demonstrating that the argumentation regarding the specific synoptic conditions patterns during the high ozone episodes could be observed at both map types.*

8) The ECMWF vertical velocity values are not presented very well. In figure 3 almost everything is white because the color scale ranges from -4 to 4. The color scale should be selected to maximize the information content that is being conveyed. The same color scale for the other vertical velocity panel in Figure 3 should be used? To effectively understand how vertical velocity changes from one day to the next the same color scale is required.

*Following the reviewer's suggestion we improved the vertical velocity scales of Fig. 3, by using the same color ranges as in Fig. 8 (-2 to 2).*

9) The surface ozone map plots have poor resolution and they have far too much empty space over Scandinavia and Russia. The focus is on the Mediterranean, please focus on this region.

*At first, we think that the presentation is improved with the new magnified Figs plotted in the best available resolution. These maps present all available measurements from rural air pollution stations at the AIRBASE database of the European Environmental Agency. We cannot avoid the empty space over Scandinavia and Russia as there are only few available stations there. The focus is on the Mediterranean, of course, but one of the*

*main points of the paper is that the ozone concentrations measured at a certain rural station are influenced by large synoptic patterns, so we need to examine the extended geographical domain. This is very well shown in the May episode (Fig. 9) where the dense measuring network in Central Europe helps to demonstrate that at the same time that we observe an ozone episode at the Western Mediterranean coast very high concentrations are observed also at Central Europe over the area of strong descending winds at the periphery of the N. European anticyclone.*

Unrelated comment: When referring to an ozone measurement of ppb, mixing ratio should be used rather than concentration.

*Although many times "concentration" is used also for describing values in ppb, we agree with the reviewer that it is more accurate to use "mixing ratio" instead, so the corresponding replacement has been made.*